# CSF total tau as a proxy of synaptic degeneration

Carolina Soares [1,2], Bruna Bellaver[1], Pamela C. L. Ferreira [1], Guilherme Povala[1], Cristiano Schaffer Aguzzoli [1,3,4], João Pedro Ferrari-Souza [1,2], Hussein Zalzale[1], Firoza Z. Lussier[1], Francieli Rohden[1,2], Sarah Abbas[1], Guilherme Bauer-Negrini[1], Douglas Teixeira Leffa [1], Andréa Lessa Benedet[5], Rebecca Langhough[6,7], Tobey J. Betthauser [6,7], Bradley T. Christian [7], Rachael E. Wilson[6,7], Dana L. Tudorascu[1,8], Pedro Rosa-Neto [9,10,11], Thomas K. Karikari [1,5], Henrik Zetterberg [5,7,12,13,14,15], Kaj Blennow [5,12], Eduardo R. Zimmer [2,4,16,17,18], Sterling C. Johnson [6,7] & Tharick A. Pascoal [1,19] ✉

Cerebrospinal fluid (CSF) total tau (t-tau) is considered a biomarker of neuronal degeneration alongside brain atrophy and fluid neurofilament light chain protein (NfL) in biomarker models of Alzheimer's disease (AD). However, previous studies show that CSF t-tau correlates strongly with synaptic dysfunction/degeneration biomarkers like neurogranin (Ng) and synaptosomal-associated protein 25 (SNAP25). Here, we compare the association between CSF t-tau and synaptic degeneration and axonal/neuronal degeneration biomarkers in cognitively unimpaired and impaired groups from two independent cohorts. We observe a stronger correlation between CSF t-tau and synaptic biomarkers than neurodegeneration biomarkers in both groups. Synaptic biomarkers explain a greater proportion of variance in CSF t-tau levels compared to neurodegeneration biomarkers. Notably, CSF t-tau levels are elevated in individuals with abnormalities only in synaptic biomarkers, but not in individuals with abnormalities only in neurodegeneration biomarkers. Our findings suggest that CSF t-tau is a closer proxy for synaptic degeneration than for axonal/neuronal degeneration.

Alzheimer's disease (AD) is pathologically characterized by deposition of amyloid-beta (Aβ) and hyperphosphorylated tau in the brain, as well as synaptic and neuronal degeneration[1,2]. Synaptic degeneration has been suggested to precede widespread neurodegeneration in pre-clinical and postmortem studies[1–4]. In vivo biomarker studies have demonstrated that synaptic degeneration, as indicated by cerebrospinal fluid (CSF) proteins such as growth-associated protein-43, synaptosomal-associated protein 25 (SNAP25) and neurogranin (Ng), precedes abnormalities in established axonal/neuronal degeneration biomarkers (such as hippocampal volume (HCV) and neurofilament light chain (NfL))[5], supporting the importance of evaluating both

synaptic and neuronal degeneration separately to better model AD progression.

The tau protein is known to play a role in stabilizing microtubules in the axon[6]. Additionally, recent experimental studies have shown that tau also plays crucial physiological roles in regulating synaptic plasticity. Specifically, synaptic activity increases tau extracellular levels, with presynaptic glutamate release driving this elevation[7]. Furthermore, tau mediates dendritic spine density and morphology and the stabilization of glutamatergic receptors in the postsynaptic compartment[8,9]. In AD, pathologic tau can be translocated from the axon to synaptic sites where it is associated with

**Table 1 | Demographic and key characteristics of participants across all cohorts**

| Characteristics | CU (n = 760) | CI (n = 932) | P-value |
|---|---|---|---|
| Age, years mean (s.d.) | 69.6 (7.5) | 72.7 (7.7) | <0.0001 |
| Females, n (%) | 465 (61 %) | 387 (42 %) | <0.0001 |
| Race | | | |
| White, n (%) | 700 (92 %) | 883 (95 %) | 0.028 |
| Unknown, n (%) | 4 (0.5 %) | 3 (0.3 %) | 0.708 |
| Ethnicity | | | |
| Not Hispanic/Latino, n (%) | 733 (96 %) | 901 (97 %) | 0.790 |
| Unknown, n (%) | 4 (0.5 %) | 3 (0.3 %) | 1.000 |
| APOEε4 carriers, n (%) | 265 (35 %) | 516 (55 %) | <0.0001 |
| Aβ-positive, n (%) | 265 (35 %) | 686 (74 %) | <0.0001 |
| HCV, z-score mean (s.d.) | −0.02 (1.0) | −1.3 (1.4) | <0.0001 |
| CSF Aβ$_{1-42}$, z-score mean (s.d.) | −0.52 (1.1) | −1.4 (0.8) | <0.0001 |
| CSF NfL, z-score mean (s.d.) | −0.002 (0.6) | 1.3 (2.5) | <0.0001 |
| CSF t-tau, z-score mean (s.d.) | 0.15 (1.2) | 1.0 (1.7) | <0.0001 |
| CSF Ng, z-score mean (s.d.) | 0.22 (1.2) | 0.95 (1.4) | <0.0001 |
| CSF SNAP25, z-score mean (s.d.) | 0.28 (1.3) | 0.91 (1.7) | <0.0001 |

Values are mean (±s.d.) for continuous variables and n (%) for categorical variables. Continuous variables were tested with a two-sided Student's t-test. Categorical variables were tested with Fisher's exact test. Missing values: APOEε4 (n = 56), HCV (n = 91), NfL (n = 1064), Ng (n = 1078), SNAP25 (n = 1406). Hippocampal volume (HCV). Amyloid-β (Aβ). Neurofilament light protein (NfL). Neurogranin (Ng). P-values refer to the comparison between CU and CI.

disruption of synaptic transmission[10–12], leading to synaptic degeneration that can occur independent of neuronal loss[12].

CSF total-tau (t-tau) has been considered a key biomarker of neuronal injury, reflecting the severity of axonal and neuronal damage[13]. Traditionally, CSF t-tau – which targets tau isoforms irrespective of their phosphorylation – has been linked to advanced neurodegeneration, along with NfL, brain atrophy, and glucose uptake in biomarker models of AD[5]. This link is further supported by biomarker studies that show a marked increase in t-tau levels corresponding to axonal damage[13,14]. Importantly, increased levels of CSF t-tau are associated with higher Aβ burden[15,16]. While not strongly correlated with neurofibrillary tangles detected by positron emission tomography (PET) imaging[17], elevated t-tau in CSF is characteristic of AD dementia, distinguishing it from many other neurodegenerative disorders, and can be used to predict accelerated clinical progression[18,19]. Current research also indicates a strong association between CSF t-tau and biomarkers that reflect synaptic degeneration, such as Ng and SNAP25, in both mild cognitive impairment (MCI) and preclinical AD, when t-tau levels have been found to triple[20–23], before substantial neuronal loss. This suggests that CSF t-tau may be a more relevant biomarker for early synaptic changes than for later-stage neurodegeneration.

In this study, we demonstrate that CSF t-tau exhibits a stronger link with classical synaptic biomarkers, CSF SNAP25 and Ng, than with classical neurodegeneration markers such as hippocampal atrophy and CSF NfL. Elevated CSF t-tau levels were observed in individuals presenting abnormalities in synaptic biomarkers alone. These findings underscore the potential of CSF t-tau to serve as a complementary indicator of synaptic dysfunction in research cohorts and clinical trial settings.

## Results

### Participants

We investigated 1692 individuals (mean age ± s.d. = 70.9 ± 7.8) from two research-based cohorts (1407 from ADNI and 285 from WRAP). Overall, there were 760 cognitively unimpaired (CU) and 932 cognitively impaired (CI) individuals, the latter group being older, more likely to be APOEε4 carrier (55 %), Aβ + (74%), and to present higher levels of CSF t-tau, Ng and SNAP25, and lower HCV and CSF Aβ42 compared to CU individuals. Demographic characteristics of the overall population and individual cohorts are presented in Table 1 and Supplementary Tables 1–2, respectively.

### CSF t-tau shows a stronger association with synaptic than neurodegeneration biomarkers

In CU individuals, the three neurodegeneration biomarkers showed low-moderate intercorrelation (HCV and NfL: r = 0.192 [0.089–0.290], Padj = 0.002; HCV and t-tau: r = 0.056 [−0.017 to 0.128], Padj = 0.949; NfL and t-tau: r = 0.472 [0.387–0.549], Padj <0.001; Fig. 1, left). In contrast, the synaptic biomarkers Ng and SNAP25 showed a high correlation (r = 0.828 [0.777–0.868], Padj <0.001; Fig. 1, left). Overall, we observed weak correlations between neurodegeneration and synaptic biomarkers. Accordingly, the correlations between HCV and Ng (r = 0.016 [−0.089 to 0.021], Padj = 1.000; Fig. 1, left) and SNAP25 (r = 0.043 [−0.099 to 0.183], Padj = 1.000; Fig. 1, left) were not significant, and the correlations between NfL and Ng (r = 0.349 [0.254–0.438], Padj <0.001; Fig. 1, left) and SNAP25 (r = 0.382 [0.255–0.497], Padj <0.001; Fig. 1, left) were moderate. On the other hand, t-tau and the synaptic biomarkers were strongly correlated (Ng: r = 0.886 [0.861–0.906], Padj <0.001, SNAP25: r = 0.868 [0.828–0.899], Padj <0.001; Fig. 1, left). Additionally, CSF Aβ42 and tau PET SUVR showed low correlations with all other biomarkers, ranging from r = −0.049 to 0.274 and r = 0.035–0.296, respectively (Fig. 1, left). All estimates are in Supplementary Table 3. Analyzing the CU group by cohort, ADNI and WRAP cohorts showed a similar pattern to the overall analysis, showing moderate to high correlations between CSF t-tau and Ng and SNAP25, while correlations with NfL and HCV were weaker (Supplementary Fig. 1 and Supplementary Tables 4–5).

Notably, comparison between associations using the confidence interval overlap revealed that in CU individuals, the magnitude of the association of t-tau with HCV (β = 1.04 [0.95–1.13], P = 0.384) and NfL (β = 1.98 [1.72–2.27], P < 0.001) were significantly lower than those of t-tau with Ng (β = 2.49 [2.37–2.62], P < 0.001) and SNAP25 (β = 2.29 [2.15–2.45], P < 0.001) (Fig. 2, left, Supplementary Fig. 2 and Supplementary Table 6). To further evaluate the specificity of the observed associations, we repeated the association analysis, substituting CSF t-tau for FDG PET. FDG PET did not show the same pattern of stronger correlations with synaptic biomarkers (Supplementary Table 7). Stratification by sex revealed a similar overall pattern, with the additional observation that females showed a stronger association between t-tau and SNAP25 compared to males (Supplementary Fig. 3 and Supplementary Table 8). For comparative analysis, we examined the associations between synaptic biomarkers and both CSF p-tau181 and plasma p-tau217 (WRAP cohort only) in individuals where these measures were concurrently available. CSF p-tau181 exhibited a statistically significant association with synaptic markers relative to CSF t-tau. Plasma p-tau217 did not show a statistically significant relationship with synaptic biomarkers in the sample analyzed (Supplementary Fig. 4 and Supplementary Table 9). In the main analysis by cohort, CU individuals from ADNI and from WRAP showed similar results when comparing the associations of t-tau and HCV, and t-tau and synaptic biomarkers. However, the association of CSF t-tau with NfL was stronger in the WRAP cohort (Supplementary Fig. 5 and Supplementary Tables 10–11).

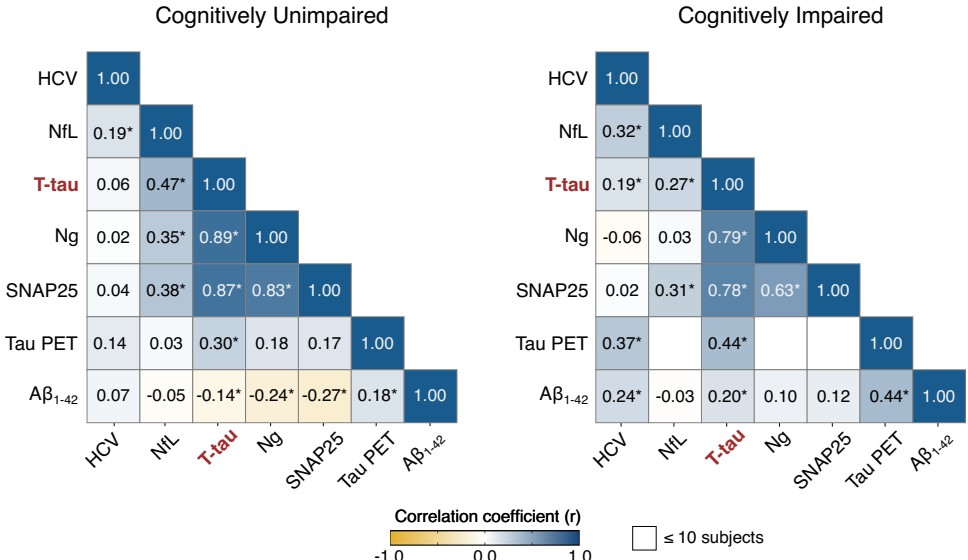

**Fig. 1 | Correlation between Alzheimer's disease biomarkers underscores the link between CSF t-tau levels and synaptic degeneration.** Correlation matrix shows significant two-sided Pearson's coefficient ($r$) correlations after correcting for multiple comparisons (*Padj-value < 0.05) among established biomarkers of neurodegeneration (HCV, CSF NfL and t-tau), synaptic (CSF Ng and SNAP25), and tau tangles (tau PET) and Aβ (CSF Aβ$_{1-42}$) pathologies in (left) cognitively unimpaired ($n = 760$) and (right) cognitively impaired ($n = 932$) individuals. All variables were log-transformed and z-scored, and HCV and CSF Aβ$_{1-42}$ were inverted to indicate higher values indicate more pathology. Data points with sample sizes of 10 or fewer are omitted and displayed in white. Amyloid-β (Aβ). CSF (cerebrospinal fluid). Total-tau (T-tau). Classical neurodegeneration biomarkers: Hippocampal volume (HCV) and Neurofilament light chain protein (NfL). Classical synaptic biomarkers: Neurogranin (Ng) and Synaptosomal-associated protein 25 (SNAP25). Positron emission tomography (PET). Source data are provided as a Source Data file.

In CI individuals, the correlations between neurodegeneration biomarkers were low-to-moderate (HCV and NfL: $r = 0.324$ [0.207–0.432], Padj <0.001; HCV and t-tau: $r = 0.189$ [0.124–0.254], Padj <0.001; NfL and t-tau: $r = 0.266$ [0.152–0.373], Padj <0.001) (Fig. 1, right). The synaptic biomarkers showed a strong correlation ($r = 0.632$ [0.489–0.742], Padj <0.001) (Fig. 1, right). In contrast, neurodegeneration biomarkers were weakly correlated with the synaptic biomarkers. Accordingly, the correlation between HCV and Ng ($r = -0.062$ [−0.188 to 0.066], Padj = 1.000) and SNAP25 ($r = 0.018$ [−0.192 to 0.7], Padj = 1.000), and between NfL and Ng ($r = 0.029$ [−0.092 to 0.149], Padj = 1.000) were not significant, and between NfL and SNAP25 ($r = 0.315$ [0.117–0.488], Padj = 0.016) (Fig. 1, right) was weak. Conversely, t-tau and synaptic biomarkers showed a strong correlation (Ng: $r = 0.792$ [0.743–0.833], Padj <0.001, SNAP25: $r = 0.776$ [0.679–0.846], Padj <0.001) (Fig. 1, right). CSF Aβ42 and tau PET SUVR exhibited a low-to-moderate correlation with all biomarkers evaluated. Correlations between tau PET and CSF NfL, Ng, or SNAP25 were omitted due to limited sample size. All estimates are described in Supplementary Table 3. Considering the low number of CI in the WRAP cohort ($n = 16$), we performed Pearson correlation tests only in ADNI. Accordingly, t-tau strongly correlated with Ng and SNAP25 but was weakly correlated with NfL and HCV (Supplementary Fig. 1 and Supplementary Table 4).

Similarly, the association of t-tau with HCV (β = 0.87 [0.82–0.92], $P < 0.001$) and NfL (β = 1.40 [1.24–1.58], $P < 0.001$) were significantly lower than those of t-tau with Ng (β = 2.67 [2.43–2.93], $P < 0.001$) and SNAP25 (β = 2.02 [1.80–2.27], $P < 0.001$) (Fig. 2, right, Supplementary Fig. 2 and Supplementary Table 12). Moreover, the ADNI cohort showed similar results to the overall analysis (Supplementary Fig. 5 and Supplementary Table 10). The FDG PET AD-ROI analysis revealed significant associations with HCV, CSF NfL, and CSF t-tau, but not with Ng or SNAP25 (Supplementary Table 13). When stratified by sex, the overall pattern of associations remained consistent. Notably, among CI, females showed a stronger association between t-tau and Ng compared to males (Supplementary Fig. 3 and Supplementary

Table 14). Subdivision of the CI group into MCI and dementia yielded similar mean estimates (Supplementary Fig. 3 and Supplementary Table 15).

## CSF t-tau variance is best explained by synaptic biomarkers
In both CU and CI groups, synaptic degeneration biomarkers explained most of the CSF t-tau variance (CU: p$R^2$ = 0.86 ± 0.02, 95.7% $R^2$; CI: p$R^2$ = 0.74 ± 0.04, 89.9% $R^2$) (Fig. 3). On the other hand, neurodegeneration biomarkers explained a small portion of the t-tau variability (CU: p$R^2$ = 0.12 ± 0.05, 15.2% $R^2$; CI: p$R^2$ = 0.17 ± 0.07, 23.8% $R^2$). Additionally, in the CU group, synaptic biomarkers better explained t-tau variance compared to neurodegeneration biomarkers (AIC = 179 and 518, respectively) (Table 2). Similarly, in the CI group, synaptic biomarkers better explained t-tau variance compared to neurodegeneration biomarkers (AIC = 163 and 259, respectively) (Table 2). Notably, in both groups, the full model best explained the t-tau variance (CU: AIC = 154; CI: 146) (Table 2). The analysis by cohort revealed similar results, showing that synaptic biomarkers explained more variance in t-tau levels than neurodegeneration in the two cohorts for both CU and CI groups (Supplementary Tables 16–17).

## Abnormal synaptic markers are linked to elevated CSF t-tau independent of neuronal degeneration
The classification of individuals in N/S groups generated an unbalanced distribution of CU and CI individuals across the four groups (Fig. 4, left). Indeed, we observed 87%, 77%, 65% and 38% of CU individuals in the N−/S−, N−/S+, N+/S−, and N+/S+ groups, respectively. Thus, we analyzed the whole population across the AD clinical continuum and compared t-tau levels across N/S groups. Accordingly, t-tau levels in the N+/S− group were lower compared to N−/S+ when using Ng (0.200 ± 1.119, T = −14.262, Padj <0.001, Fig. 4, right) as a proxy for S. Additionally, N−/S+ individuals did not show differences in t-tau levels compared to N+/S+ (1.123 ± 1.104, T = 1.171, Padj = 0.648). CSF t-tau levels did not differ between N−/S− and N+S− individuals (0.912 ± 1.133, T = −0.739, Padj = 0.882). Moreover, all comparisons

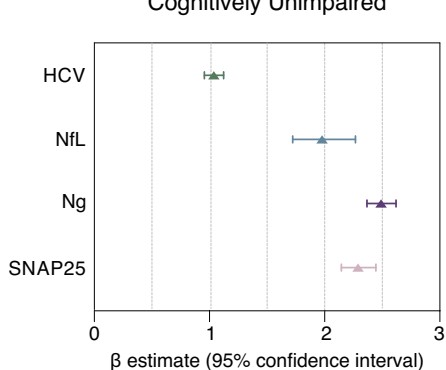
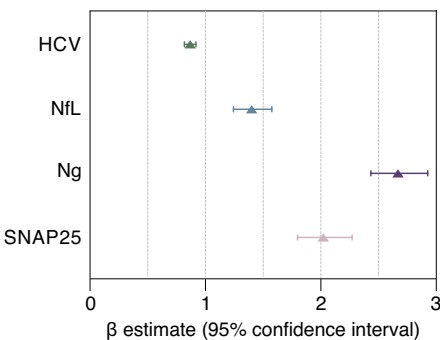

**Fig. 2 | CSF t-tau is more strongly associated with synaptic than neurodegeneration biomarkers.** Mean β estimates with 95% confidence intervals from linear regressions, adjusted for age, sex, and cohort, showing the association of CSF t-tau with biomarkers of neurodegeneration and synaptic degeneration in (left) cognitively unimpaired ($n = 760$) and (right) cognitively impaired ($n = 932$) individuals. Non-overlapping confidence intervals indicate statistically significant differences.

All variables were log-transformed and z-scored. HCV was inverted, so higher values indicate more pathology. Cerebrospinal fluid (CSF). Total-tau (t-tau). Classical neurodegeneration biomarkers: Hippocampal volume (HCV) and CSF Neurofilament light chain protein (NfL). Classical synaptic biomarkers: CSF Neurogranin (Ng) and Synaptosomal-associated protein 25 (SNAP25). Source data are provided as a Source Data file.

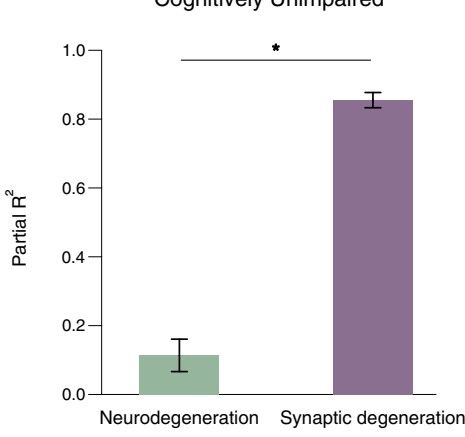
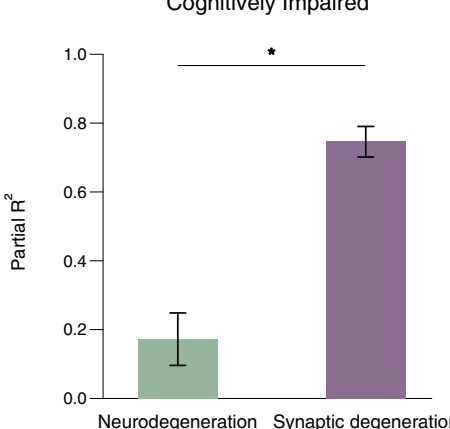

**Fig. 3 | Synaptic biomarkers explain a greater proportion of variance in CSF t-tau than neurodegeneration biomarkers.** Bar charts show mean (±s.d.) the partial $R^2$ in (left) cognitively unimpaired ($n = 188$) and (right) cognitively unimpaired groups ($n = 84$). Partial $R^2$ values were estimated only for individuals who had complete data for all the following biomarkers: CSF t-tau as the outcome and biomarkers reflecting neurodegeneration (HCV and CSF NfL) and synaptic degeneration (CSF Ng and SNAP25) as predictors in linear regression models. Models were adjusted for age, sex, and cohort, and compared using AIC. Bootstrapping

($n = 1000$ iterations) was applied to assess the stability of the partial $R^2$ values. *indicate Akaike Information Criterion difference (ΔAIC) between neurodegeneration and synaptic degeneration >15. Cognitively unimpaired ΔAIC = 339; cognitively impaired ΔAIC = 95. Cerebrospinal fluid (CSF). Total-tau (t-tau). Hippocampal volume (HCV). Neurofilament light chain protein (NfL). Neurogranin (Ng). Synaptosomal-associated protein 25 (SNAP25). Standard deviation (s.d.). Source data are provided as a Source Data file.

remained significant after adjusting for Aβ burden. Results were replicated when using SNAP25 as a proxy for synaptic degeneration and partially replicated when using CSF NfL as a proxy for neurodegeneration. In both cases, synaptic abnormalities remain more strongly associated with elevated CSF t-tau levels compared to neurodegeneration alone. Notably, the N+/S+ group, defined using CSF NfL, showed even higher t-tau levels compared to the N−/S+ group (Supplementary Fig. 6). Complete model estimates can be found in Supplementary Tables 18–21. Individual cohort analysis showed consistent significant group differences (Supplementary Tables 22–25).

## Discussion

In this study, we investigated the association of CSF t-tau with markers of synaptic and axonal/neuronal degeneration across the aging and the AD spectrum in two independent large cohorts. Our findings suggest that CSF t-tau was more closely associated with classical biomarkers of synaptic degeneration than those of neuronal degeneration. Notably,

we found evidence that abnormalities in biomarkers of synaptic degeneration, even in the absence of neuronal degeneration abnormality indexed by HCV, were strongly associated with increased CSF t-tau levels. Thus, CSF t-tau shows the potential to be used as a proxy of synaptic degeneration.

We found strong associations between CSF t-tau with CSF Ng and SNAP25 in both CU and CI individuals, reinforcing mounting evidence showing a strong link between CSF t-tau and Ng[15,16,24–27] SNAP25[24,25], GAP43[24,25], synaptotagmin-1[25], and neuropentraxin-2[24]. In CI individuals, we observed a modest association between CSF t-tau and HCV, CSF NfL and FDG PET, while in CU individuals, CSF t-tau was only moderately associated with CSF NfL. These results align with prior studies in CI individuals showing low-moderate associations with FDG PET[28], HCV, and cortical thickness[28–30] and plasma NfL[30], but differ from reports that found significant associations in the CU population[31] or observed stronger correlations in a CU and CI combined sample[32,33]. Given the limited evidence in CU populations and variability across

**Table 2 | Proportion of variation of CSF t-tau biomarker levels explained by biomarkers of synaptic degeneration and neurodegeneration**

| Model | pR² | pR² (%) | R² | ΔpR² | AIC |
|---|---|---|---|---|---|
| **Cognitively unimpaired** | | | | | |
| Full | - | - | 0.90 | - | 154[a,b] |
| Neurodegeneration | 0.12 ± 0.05 | 15.0% | 0.26 | 0.52 | 518[b] |
| Synapse degeneration | 0.86 ± 0.02 | 95.3% | 0.78 | −0.52 | 179 |
| **Cognitively impaired** | | | | | |
| Full | - | - | 0.82 | - | 146[a,b] |
| Neurodegeneration | 0.17 ± 0.07 | 23.6% | 0.29 | 0.48 | 258[b] |
| Synapse degeneration | 0.74 ± 0.04 | 90.3% | 0.77 | −0.48 | 163 |

Proportion of variation of CSF t-tau levels explained by neurodegeneration (HCV and CSF NfL combined) and synaptic degeneration (Ng and SNAP25 combined) were estimated using partial $R^2$ (p$R^2$) from multivariable linear regression models adjusted for age, sex and cohort, only for individuals with complete data for all five biomarkers, in cognitively unimpaired ($n = 188$) and impaired ($n = 84$) groups. p$R^2$ values are reported as mean ± s.d. obtained by bootstrapping ($n = 1000$ iterations). Percentual p$R^2$ was calculated as the p$R^2$ of each model divided by the total $R^2$ of the model (100* p$R^2$/$R^2$). The percentage of p$R^2$ does not sum to 100% due to shared variability. p$R^2$ models were calculated based on all variables (full model: t-tau ~HCV + NfL + Ng + SNAP25); only biomarkers of neurodegeneration (synaptic degeneration model: t-tau ~HCV + NfL), or only biomarkers of synaptic degeneration (neurodegeneration model: t-tau ~Ng + SNAP25). Δp$R^2$ indicates the difference between neurodegeneration and synaptic degeneration p$R^2$ values. AIC values were calculated for models assessing t-tau levels associated with biomarkers of neurodegeneration and synaptic degeneration.
*AIC* Akaike information criterion. *CSF* cerebrospinal fluid. *t-tau* Total-tau. *HCV* hippocampal volume (HCV). *NfL* neurofilament light chain protein. *Ng* Neurogranin. *SNAP25* synaptosomal-associated protein 25. Source data are provided as a Source Data file.
[a]AIC > 15 compared with neurodegeneration.
[b]AIC > 15 compared with synaptic degeneration.

studies, further research is needed to clarify t-tau's role in preclinical stages. Altogether, these findings suggest that CSF t-tau is well associated with synaptic biomarkers before cognitive impairment, but only moderately - and most frequently only in symptomatic individuals - with measures of atrophy, neuronal injury, and glucose uptake.

CSF t-tau has been commonly used as a neuronal degeneration biomarker in observational and interventional AD studies[5,34]. This discrepancy motivated our head-to-head investigation and direct comparison of synaptic and neuronal degeneration biomarkers with CSF t-tau. In both CU and CI individuals, we found that CSF t-tau was more strongly associated with synaptic biomarkers, and with most of its variance explained by these markers, suggesting that CSF t-tau better reflects related synaptic changes[35]. Our findings also support the notion that this link between CSF t-tau and synaptic degeneration emerges before cognitive impairment and persists throughout the AD continuum, consistent with previous literature showing elevated synaptic biomarkers in preclinical AD[25]. Stratified analysis in MCI and AD groups further confirmed this pattern, underscoring the strong link between CSF t-tau and synaptic biomarkers across the clinical stages of AD as shown previously[24]. Therefore, our study not only reaffirms the strong link between CSF t-tau and synaptic biomarkers but also indicates that CSF t-tau is more closely associated with synaptic degeneration than neurodegeneration, offering a different interpretation to its abnormality in AD progression. This carries significant implications for research cohorts and clinical trials, especially in settings where CSF t-tau is available but is underutilized as a neurodegeneration marker due to the availability of structural MRI and NfL. Specifically, CSF t-tau shows promise for exploring the relationship between synaptic degeneration and AD, as well as monitoring the impact of drugs on synaptic dysfunction.

Furthermore, we found that females exhibit a stronger association of CSF t-tau with SNAP25 in the CU group, and with Ng in the CI group. Although one study found no sex differences in SNAP25 levels in Aβ-positive CU individuals, data on this biomarker remain limited[25]. Alternatively, different groups have reported increased Ng in females, both in combined CU and CI samples, regardless of Aβ status[15], and in a sample stratified by Aβ status[26], and specifically in CU Aβ-positive individuals[25]. These findings indicate that CSF t-tau is more strongly linked to synaptic biomarkers in females, starting from the early stages of AD.

When classifying individuals based on their synaptic and neuronal degeneration status, we showed that abnormalities in synaptic biomarkers alone were associated with increased CSF t-tau levels, particularly when HCV was used as a proxy for neuronal degeneration. The hippocampus is one of the earliest brain regions affected in the neurodegenerative process of AD[35], and reduced HCV has been strongly linked to disease progression[36]. Our findings indicate that despite this early vulnerability, synaptic dysfunction shows a stronger and more immediate association with CSF t-tau levels than hippocampal atrophy. The observation that CSF NfL abnormalities alone were associated with elevated CSF t-tau − albeit less strongly than when accompanied by synaptic abnormalities − suggests that CSF NfL may serve as an earlier neurodegeneration biomarker, potentially overlapping with synaptic dysfunction. This adds to the growing evidence base documenting inconsistencies among neurodegeneration markers[31,33]. Moreover, animal studies have demonstrated that tau protein can be associated with synaptic degeneration independently of neuronal death. For example, pathogenic tau binds to presynaptic vesicles, causing reduced vesicle mobility, decreased release rate, and impaired neurotransmission[8,12,37] - independent of neuronal loss[4,12], although more experimental studies are needed to elucidate CSF t-tau levels in this context. These findings are consistent with postmortem and biomarker studies showing that synaptic dysfunction precedes neuronal loss[2,38–40], and altogether, these results underscore the need for further studies to better understand the mechanisms underlying CSF t-tau dynamics and its association with AD progression and refine its biological interpretation.

We found an association between CSF t-tau and CSF Aβ42, further supporting the notion that CSF t-tau may be sensitive to AD pathology[18,19]. However, the modest associations of CSF t-tau with tau PET highlight that it is not a strong proxy for tau tangle pathology[17,41]. Our findings also suggest that higher levels of CSF t-tau, Ng, and SNAP25 are associated with lower Aβ burden in CU individuals, consistent with some studies[16,23], but not all[24]. Collectively, these findings suggest that CSF t-tau primarily reflects AD-related synaptic degeneration but provide only limited insights into neuronal degeneration as measured by classical biomarkers[5,42].

The strengths of our study include replicated results in two independent cohorts, a large sample size, and consistent biomarker methods across cohorts. However, our findings must be carefully interpreted due to the complex and intertwined relationship between synaptic and neuronal degeneration. In vitro studies are necessary to accurately disentangle synaptic from axonal/neuronal degeneration. The shared variance between these variables limits the statistical ability to fully separate the contributions of each pathological process. HCV limits the definition of neurodegeneration to a single region, potentially underestimating atrophy occurring in other brain regions. While thresholds are valuable research tools, they are inherently influenced by the methods and assumptions used. Additionally, the WRAP cohort is mainly composed of CU and MCI individuals, and like the currently available ADNI cohort, is also predominantly composed of White participants. It would be desirable to replicate these results in a more diverse general population.

In conclusion, our findings suggest repositioning CSF t-tau as a more specific marker of synaptic degeneration. This can potentially provide valuable insights for research cohorts and clinical trials that have this measurement available but often underutilize it.

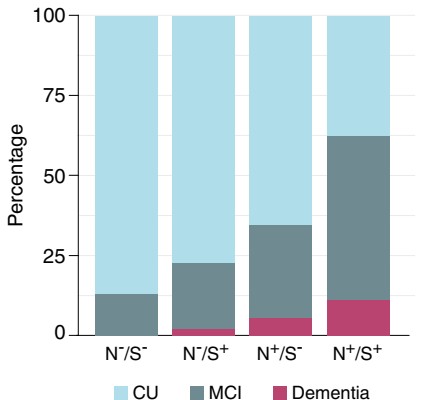

**a**  Distribution of diagnosis across N/S groups

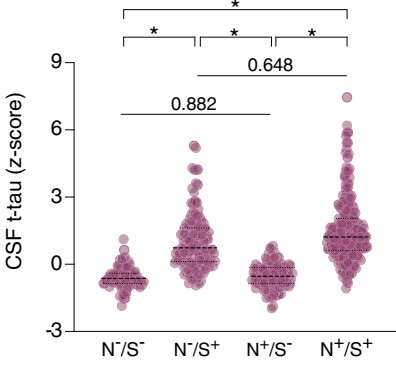

**b**  CSF t-tau levels across N/S groups

**Fig. 4 | CSF t-tau is increased in individuals with abnormal synaptic degeneration regardless of the concomitant presence of neurodegeneration. a** Bar graphs show the distribution of CU, MCI, and dementia across synaptic degeneration (S) and neurodegeneration (N) groups. N positivity was based on HCV, and S positivity was based on CSF Ng (N⁻S⁻: $n = 85$; N⁻S⁺: $n = 137$; N⁺S⁻: $n = 110$; S⁺N⁺: $n = 252$). Cutoffs for biomarker positivity were calculated anchored in the CU Aβ⁻ individuals. **b** Violin plots show CSF t-tau levels in individuals with abnormal S and/ or N in the whole population (CU and CI). The median is shown by the middle dashed line, and the quartiles by the top and bottom dashed lines. CSF t-tau levels were compared across groups using a linear regression model with dummy variables, adjusted for age, sex, cognitive status, amyloid burden, and cohort. Pairwise comparisons were corrected for multiple testing using Tukey's method. * Padj-value < 0.001. Amyloid-β (Aβ). Cognitively unimpaired (CU). Mild cognitive impairment (MCI). Neurodegeneration (N). Neurogranin (Ng). Synaptic degeneration (S). Cerebrospinal fluid (CSF). Total-tau (t-tau). Hippocampal volume (HCV). Neurofilament light chain protein (NfL). Neurogranin (Ng). Synaptosomal-associated protein 25 (SNAP25). Source data are provided as a Source Data file.

## Methods

### Study population

This study was conducted in accordance with the Declaration of Helsinki and applicable federal regulations and complied with all applicable ethical guidelines. Institutional Review Board (IRB) approval was not required at our site, as all clinical data were obtained in a de-identified format from external study centers that adhere to established ethical standards. We studied cognitively unimpaired (CU), MCI and individuals with dementia due to AD, obtained from two cohorts: the Alzheimer's disease Neuroimaging Initiative (ADNI) database (adni.loni.usc.edu) a multi-center research effort initiated in 2004, and the Wisconsin Registry for Alzheimer's Prevention study (WRAP), based in Wisconsin, USA (https://wrap.wisc.edu). The overall ADNI study received approval from the IRBs of all participating sites, and all individuals provided informed consent. In agreement with ADNI policies, this study's principal investigator signed the ADNI Data Use Agreement and is authorized to access and use ADNI data. All study protocols for the WRAP cohort were approved by the IRB at the University of Wisconsin–Madison (IRB: 2023-1522, approval date 1/23/2024), and all participants provided signed informed consent before participation. The ADNI database provides clinical assessments, neuroimaging, and biomarker measurements. The data for this study were collected from ADNI phases ADNI1, ADNIGO, ADNI2, and ADNI3, with the most recent data downloaded in April 2024. The WRAP cohort included participants with clinical evaluation and multiple cognitive assessments, neuroimaging, and biomarker measurements. Participants were excluded if they were not fluent in English, lacked the necessary visual or auditory acuity for neuropsychological testing, or were not in good health. Specifically, individuals with any diseases expected to interfere with their ability to participate over time were not included.

The cognitive groups were defined by either clinical diagnosis (for WRAP) or by Clinical Dementia Rating (CDR) (for ADNI): CU individuals had a CDR = 0 and no objective cognitive impairment, MCI individuals had a CDR of 0.5, and AD dementia had a CDR ≥0.5[43]. Participants included were >50 years old and had a clinical diagnosis or CDR, CSF t-tau, CSF Aβ42, and at least one of the following biomarkers: HCV, CSF NfL, CSF Ng, CSF SNAP25, tau PET. An additional subset from the ADNI cohort with available glucose metabolism data measured by [¹⁸F] fluorodeoxyglucose (FDG)-PET was also included.

### Fluid biomarkers

CSF procedures have been detailed for the two cohorts[44,45]. Aβ42 and t-tau were measured by fully automated Elecsys® assays (Roche Diagnostics). In the ADNI cohort, Ng was quantified by an in-house immunoassay[46] and NfL levels were determined through a commercial ELISA (Uman Diagnostics, Umea, Sweden)[47] performed at the Clinical Neurochemistry Laboratory, Sahlgrenska University Hospital. In the WRAP cohort, both Ng and NfL were measured with the NeuroToolKit (NTK) and a Cobas 411 analyzer[48]. SNAP25 was determined similarly across cohorts with a commercially available assay (Quanterix Simoa SNAP25 Advantage Kit), which targets the soluble N-terminal fragment SNAP25 (aa 2-47) using a monoclonal mouse antibody against the full-length unaltered SNAP25 protein as the capture antibody. CSF measures from ADNI, with notes indicating potential analytical problems (e.g., above limit of quantification), were removed from analysis. Participants were classified as Aβ-positive using CSF Aβ42 below 976.6 pg/ml (ADNI)[49] and CSF Aβ42/Aβ40 below 0.046 (WRAP)[48]. In WRAP, we used the CSF Aβ42/Aβ40 ratio to leverage the published cutoff[48]. We used CSF Aβ42 alone in ADNI to preserve sample size due to the limited availability of concomitant synaptic biomarkers and Aβ40 data in the table "UPENNBIOMK_ROCHE_ELECSYS.csv". Plasma p-tau217 was measured in the WRAP cohort with the ALZpath assay on the Simoa® single-molecule array platform[50].

### Magnetic resonance imaging/PET biomarkers

Detailed imaging protocols for ADNI and WRAP were described elsewhere[50,51]. HCV was determined using the Freesurfer software package[52] and adjusted by Freesurfer version, and by intracranial volume[53] from CU Aβ-negative individuals at baseline according to published protocol[54]. ADNI tau PET was quantified using the tracer Flortaucipir, and WRAP tau PET was quantified using MK-6240[55,56]. Standard Uptake Values (SUVR) from meta-temporal-ROI comprised of values from regions such as the bilateral entorhinal, amygdala, fusiform, inferior, and middle temporal cortices[57]. ADNI glucose uptake was measured with [¹⁸F]FDG PET, and the SUVR from the AD-ROI

comprised values from the regions bilateral angular, bilateral posterior cingulate, and bilateral inferior temporal gyri[58].

## Statistical analysis

Differences between CU and CI individuals within cohorts were assessed with an unpaired two-sided $t$-test for continuous variables (age, years of education, HCV, Aβ42, CSF t-tau, tau PET, CSF NfL, CSF Ng, CSF SNAP25) and $\chi^2$ for categorical variables (sex, APOEε4, Aβ status, race, and ethnicity). Correlations between biomarkers were estimated with Pearson's correlation with a Bonferroni test to adjust for multiple comparisons, and results are expressed as the r coefficient, 95% confidence interval, and Padj-value. Linear regression models adjusted for age, sex, and cohort were used to test associations between biomarkers and are expressed as β coefficient, 95% confidence interval, and $P$-value. Additionally, we conducted the same analysis in females and males separately, as well as in individuals with MCI and dementia separately. Moreover, we used the confidence intervals to compare differences between models. We estimated the proportion of t-tau variance explained by neurodegeneration (HCV and NfL) and synaptic degeneration (Ng and SNAP25) using the *rsq* package. We calculated the partial coefficient of determination ($pR^2$) and the percentage of the total $R^2$ of the model[59], using only participants with all four biomarkers available. Briefly, the full model included t-tau as the outcome, and both synaptic and neurodegeneration biomarkers as predictors. The partial models included either synaptic or neurodegeneration biomarkers as predictors. The linear regressions used to calculate the $pR^2$ were adjusted by age, sex, and cohort. Bootstrapping ($n = 1000$ iterations) was applied to assess the stability of the $pR^2$ values, yielding the reported mean $pR^2$ and standard deviation (s.d.). The model that best explained t-tau variance was determined by comparing $pR^2$ values using the Akaike Information Criterion (AIC; lower value indicates better fit, and AIC difference >15 was considered significant)[60,61]. Additionally, within each cohort, four groups were generated to compare mean levels of t-tau according to their status of synaptic degeneration (S) and neurodegeneration (N): $N^-/S^-$, $N^-/S^+$, $N^+/S^-$, and $N^+/S^+$. HCV was used as a proxy for N, and either Ng or SNAP25 was used as a proxy for S. Participants were considered $N^+$ based on either HCV values below the median or CSF NfL above the median from CU Aβ-negative individuals. Participants were considered $S^+$ based on either Ng or SNAP25 values above the median from CU Aβ-negative individuals[39]. Linear regression models using N/S groups as dummy variables were employed to compare CSF t-tau mean levels between groups, adjusted for age, sex, cognitive status, cohort, and Aβ burden (CSF Aβ42), corrected for multiple comparisons with Tukey's method, and expressed as mean ± standard error (SE), T-value, and Padj-value. CSF Aβ40 and tau PET were available only in a limited subset of the ADNI cohort included in this study, and thus not used as covariates in the linear regression models. Except for demographic comparison, all tests were carried out with variables log-transformed and z-scored anchored in the CU Aβ-negative group, within each cohort. Moreover, HCV and CSF Aβ42 values were inverted; thus, higher values mean more atrophy and Aβ burden, respectively, and the β estimates were exponentiated (^−1) to reverse the log-transformation, returning values to the normal scale. Assumptions were validated through standard diagnostic checks, including histograms of the residuals, Q–Q plots, and scatterplots of residuals versus fitted values. These checks confirmed that the assumptions were met and deemed reasonable. Demographic and unadjusted correlation statistical analyses were conducted using GraphPad Prism v.9, while the other statistical tests were performed using R-Studio Statistical software package 4.2.2.

## Reporting summary

Further information on research design is available in the Nature Portfolio Reporting Summary linked to this article.

## Data availability

Inquiries regarding raw and analyzed data, as well as materials, can be directed to the corresponding author (T.A.P.). The investigators and affiliated institutions will promptly assess whether there are any intellectual property or confidentiality obligations associated with the request and will respond within one month. Data from the ADNI cohort can be accessed from https://ida.loni.usc.edu. De-identified data from the WRAP cohort will be made accessible to qualified academic researchers upon request, specifically for the purpose of replicating the methods and findings outlined in this paper. More information from the WRAP cohort can be accessed from https://wrap.wisc.edu/. Any releasable data and materials will be provided under a material transfer agreement. Please note that certain information is not publicly accessible to safeguard the privacy of the research participants. Source data is provided as a source data file. Source data are provided with this paper.

## Code availability

Results from this study were generated using R code, publicly accessible on https://github.com/PascoalLab/soares2025-ttau. All codes used open-source R packages.

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

## Acknowledgements

We thank all participants and their families for their time, data, and samples. We would also like to thank the following funding agencies that supported this work: NIH, National Institute of Aging (grants 5R01AG075336 and 5R01AG073267 to T.A.P.; P01 AG025204-17 to B.B.); Alzheimer's Association (grants AARFD-22-974627 to B.B.; AACSF-20-648075 to T.A.P.; AARFD-22-923814 to P.C.L.F.; AARGD-21-850670 to E.R.Z.; 24AARFD-1243899 to G.P.; 24AACSF-1200375 to C.S.A.; AARFD-23-1150249 to G.B.-N.; ADSF-21-831376-C, ADSF-21-831381-C and ADSF-21-831377-C to H.Z.); Alzheimer's Association, Global Brain Health Institute, and Alzheimer's Society (GBHI-ALZ-UK-23-971089 to C.S.A.); CAPES (grant 88887.696202/2022-00 to C.S.; and 88887.951210/2024-00 to C.S.A.); CNPq (grant 200691/2021-0 to J.P.F.-S.; 141357/2020-7 to C.S.); Fonds de Recherche du Québec—Santé (FRQS; Chercheur Boursier, grant 2020-VICO-279314 to P.R.-N.); CIHR-CCNA Canadian Consortium of Neurodegeneration in Aging (grants MOP-11-51-31; RFN 152985, 159815, 162303 to P.R.-N.); Weston Brain Institute (grants 8400707, 8401154 and 8401103 to P.R.-N.); Colin Adair Charitable Foundation (grant to P.R.-N.); Wallenberg Scholar (grant 2022-01018 to H.Z.; 2017-00915 and 2022-00732 to K.B.); Swedish Alzheimer Foundation (Alzheimerfonden; grant AF-930351, AF-939721 and AF-968270 to K.B.); European Union's Horizon Europe research and innovation program (grant 101053962 to H.Z.); Swedish State Support for Clinical Research (grant ALFGBG-71320 to H.Z.); Alzheimer Drug Discovery Foundation (ADDF; grant 201809-2016862 to H.Z.); Bluefield Project, the Olav Thon Foundation, the Erling-Persson Family Foundation, Stiftelsen för Gamla Tjänarinnor, Hjärnfonden, Sweden (grant FO2022-0270 to H.Z.); European Union's Horizon 2020 research and innovation program under the Marie Skłodowska-Curie (grant 860197 (MIRIADE) to H.Z.); the European Union Joint Program—Neurodegenerative Disease Research (grant JPND2021-00694 to H.Z. and JPND2019- 466-236 to K.B.); The UK Dementia Research Institute at UCL (grant Article https://doi.org/10.1038/s41591-023-02380-x UKDRI-1003 to H.Z.); National Academy of Neuropsychology (grant ALZ-NAN-22-928381 to E.R.Z.); Fundação de Amparo a pesquisa do Rio Grande do Sul (FAPERGS; grant 21/2551-0000673-0 to E.R.Z.); Instituto Serrapilheira (grant Serra-1912-31365 to E.R.Z.); Hjärnfonden (grants FO2017-0243 and ALZ2022-0006 to K.B.); The Swedish state under the agreement between the Swedish government and the County Councils, the ALF agreement (grants ALFGBG-715986 and ALFGBG-965240 to K.B.); The Alzheimer's Association 2021 Zenith Award (grant ZEN-21-848495 to K.B.). The WRAP study was supported by NIA grants AG027161, AG021155, and AG062715. T.K.K. and the Karikari Laboratory were supported by NIH/NIA (R01 AG083874, U24AG082930, P30 AG066468, RF1 AG077474, R01 AG083156, R37 AG023651, R01 AG025516, R01 AG073267, R01 AG075336, R01 AG072641, P01 AG025204), NIH/NINDS (U01 NS131740, U01 NS141777), NIH/NIMH (R01 MH108509), Aging Mind Foundation (DAF2255207), DoD (HT94252320064), the Anbridge Charitable Fund, and a professorial endowment from the Department of Psychiatry, University of Pittsburgh. The content of this article is solely the responsibility of the authors and does not necessarily represent the official views of the funders.

## Author contributions

Conceived and designed the study: Carolina Soares (C.S.), Bruna Bellaver (B.B.), Tharick A. Pascoal (T.A.P.). Contributed to the acquisition and processing of biomarker data: Guilherme Povala (G.P.), Firoza Z Lussier (F.Z.L.), Guilherme Bauer-Negrini (G.B.-N.), Andréa Lessa Benedet (A.L.B.), Rebecca Langhough Koscik (R.L.K.), Tobey J. Betthauser (T.J.B.), Bradley T. Christian (B.T.C.), Rachael E. Wilson (R.E.W.), Pedro Rosa-Neto (P.R-N.), Thomas K. Karikari (T.K.K.), Henrik Zetterberg (H.Z.), Kaj Blennow (K.B.), Sterling C. Johnson (S.C.J.). Performed statistical analysis: C.S., B.B., Dana L. Tudorascu (D.L.T.). Contributed to data processing and analysis support: Pamela C.L. Ferreira (P.C.L.F.), Cristiano Schaffer Aguzzoli (C.S.A.), João Pedro Ferrari-Souza (J.P.F-S), Hussein Zalzale (H.ZAL.), Francieli Rohden (F.R.), Sarah Abbas (S.A.), Douglas Teixeira Leffa (D.T.L.). Supervised the project and interpretation of results: B.B., P.C.L.F., Eduardo R. Zimmer (E.R.Z.), T.A.P. Drafted the initial version of the manuscript: C.S., B.B., P.C.L.F., T.A.P. All authors contributed to the interpretation of findings, revised the manuscript critically for important intellectual content, and approved the final version for submission.

## Competing interests

K.B. has served as a consultant and on advisory boards for AbbVie, AC Immune, ALZPath, AriBio, BioArctic, Biogen, Eisai, Lilly, and Moleac Pte. Ltd, Neurimmune, Novartis, Ono Pharma, Prothena, Roche Diagnostics, and Siemens Healthineers; has served at data monitoring committees for Julius Clinical and Novartis; has given lectures, produced educational materials and participated in educational programs for AC Immune, Biogen, Celdara Medical, Eisai and Roche Diagnostics, and is a co-founder of Brain Biomarker Solutions in Gothenburg AB (BBS), which is a part of the GU Ventures Incubator Program, all unrelated to the work presented in this paper. H.Z. has served on scientific advisory boards and/or as a consultant for AbbVie, Alector, ALZPath, Annexon, Apellis, Artery Therapeutics, AZTherapies, CogRx, Denali, Eisai, Nervgen, Novo Nordisk, Pinteon Therapeutics, Red Abbey Labs, reMYND, Passage Bio, Roche, Samumed, Siemens Healthineers, Triplet Therapeutics, and Wave, has given lectures in symposia sponsored by Cellectricon, Fujirebio, Alzecure, Biogen, and Roche, and is a co-founder of BBS. E.R.Z. has served on the scientific advisory board, as a consultant or speaker for Next Innovative Therapeutics (Nintx), Novo Nordisk, Biogen, Lilly, Magdalena Biosciences, and masima. He is also a co-founder and minority shareholder of Masima. P.R.-N. has served on scientific advisory boards and/or as a consultant for Eisai, Novo Nordisk, and Roche. GE HealthCare holds a license agreement with the University of Pittsburgh based on the PiB PET technology described in this paper. GE HealthCare provided no grant support for this study and had no role in the design or interpretation of results or preparation of this paper. S.C.J. serves on advisory boards for AlzPATH, Lilly, Merck, Alamar, and Enigma Biomedical. B.T.C. is a scientific advisor for Alnyham and has received equipment from Lantheus. T.K.K. has consulted for Quanterix Corporation, SpearBio Inc., Neurogen Biomarking LLC., and Alzheon, and has served on advisory boards for Siemens Healthineers and Neurogen Biomarking LLC., outside the submitted work. He has received in-kind research support from Janssen Research Laboratories, SpearBio Inc., and Alamar Biosciences, as well as meeting travel support from the Alzheimer's Association and Neurogen Biomarking LLC., outside the submitted work. T.K.K. has received royalties from Bioventix for the transfer of specific antibodies and assays to third-party organizations. He has received honoraria for speaker/grant review engagements from the NIH, UPENN, UW-Madison, the Cherry Blossom symposium, the HABS-HD/ADNI4 Health Enhancement Scientific Program, Advent Health Translational Research Institute, Brain Health conference, Barcelona-Pittsburgh conference, the International Neuropsychological Society, the Icahn School of Medicine at Mount Sinai and the Quebec Center for Drug Discovery, Canada, all outside of the submitted work. T.K.K. is an inventor on several patents and provisional patents regarding biofluid biomarker methods, targets, and reagents/compositions, that may generate income for the institution and/or self should they be licensed and/or transferred to another organization. These include WO2020193500A1: Use of a ps396 assay to diagnose tauopathies; US 63/679,361: Methods to Evaluate Early-Stage Pre-Tangle TAU Aggregates and Treatment of Alzheimer's Disease Patients; US 63/672,952: Method for the Quantification of Plasma Amyloid-Beta Biomarkers in Alzheimer's Disease; US 63/693,956: Anti-tau Protein Antigen Binding Reagents; and 2450702-2: Detection of oligomeric tau and soluble tau aggregates. The other authors declare no competing interests.

## Additional information

[1]Department of Psychiatry, School of Medicine, University of Pittsburgh, Pittsburgh, PA, USA. [2]Graduate Program in Biological Sciences: Biochemistry, Universidade Federal do Rio Grande do Sul, Porto Alegre, Brazil. [3]Global Brain Health Institute, Memory and Aging Center, University of California, San Francisco, CA, USA. [4]Brain Institute, PUCRS, Porto Alegre, Brazil. [5]Department of Psychiatry and Neurochemistry, Institute of Neuroscience and Physiology, The Sahlgrenska Academy, University of Gothenburg, Mölndal, Sweden. [6]Wisconsin Alzheimer's Institute, School of Medicine and Public Health, University of Wisconsin, Madison, WI, USA. [7]Wisconsin Alzheimer's Disease Research Center, School of Medicine and Public Health, University of Wisconsin, Madison, WI, USA. [8]Department of Biostatistics, School of Medicine, University of Pittsburgh, Pittsburgh, PA, USA. [9]Translational Neuroimaging Laboratory, McGill University Research Centre for Studies in Aging, Alzheimer's Disease Research Unit, Douglas Research Institute, Le Centre intégré universitaire de santé et de services sociaux (CIUSSS) de l'Ouest-de-l'Île-de-Montréal, Montreal, QC, Canada. [10]Department of Neurology and Neurosurgery, McGill University, Montreal, QC, Canada. [11]Brain Imaging Centre, Montreal Neurological Institute-Hospital, Montreal, QC, Canada. [12]Clinical Neurochemistry Laboratory, Sahlgrenska University Hospital, Mölndal, Sweden. [13]Department of Neurodegenerative Disease, University College London Queen Square Institute of Neurology, London, UK. [14]UK Dementia Research Institute at University College London, London, UK. [15]Hong Kong Center for Neurodegenerative Diseases, Hong Kong, China. [16]Hospital Moinhos de Vento, Porto Alegre, Brazil. [17]Department of Pharmacology, Universidade Federal do Rio Grande do Sul, Porto Alegre, Brazil. [18]Graduate Program in Biological Sciences: Pharmacology and Therapeutics, Universidade Federal do Rio Grande do Sul, Porto Alegre, Brazil. [19]Department of Neurology, School of Medicine, University of Pittsburgh, Pittsburgh, PA, USA. ✉e-mail: pascoalt@upmc.edu

