## [Transparent Peer Review file · Nature Communications]

CSF TOTAL TAU AS A PROXY OF SYNAPTIC DEGENERATION

Corresponding Author: Dr Tharick Pascoal

Version 0:

Reviewer comments:

Reviewer #1

(Remarks to the Author)

This manuscript by Soares et al., studied associations between CSF t-tau and synaptic degeneration (i.e., Ng and SANP25), as well as neurodegeneration markers (i.e., hippocampal volume and NfL) in 760 cognitively unimpaired and 932 cognitively impaired participants. Data from two independent cohorts were used to demonstrate that CSF t-tau showed higher linear correlations with synaptic degeneration markers (quantified by CSF Ng and SANP25) than with neurodegeneration markers (quantified by hippocampal volume and NfL). This is an interesting paper using reasonable approaches to address timely questions in the AD field, which largely indicates experience and competence among the authors. Overall, this study is well-designed, and the manuscript is well written. I only have a few minor concerns on the methods and results interpretation, which could be addressed in a revision.

1. Although the CSF measures were orthogonalized with respect to age and sex, it is still possible that sex-specific multivariate effects may influence the associations. To address this, could the authors consider reanalyzing the associations stratified by sex (i.e., in females and males separately)?
2. Were the reported P-values corrected for multiple comparisons using methods such as False Discovery Rate (FDR) or Bonferroni correction? If not, it would be important to acknowledge this as a limitation in the manuscript.
3. In the Statistical Analysis section, please specify which “four groups” were used to compare mean levels of t-tau. While this becomes clear in Figure 4, it should be explicitly described in the Methods section for clarity.
4. Could the authors consider visualizing the correlation results using scatter plots? While correlation coefficients (e.g., $r = 0.8$) are informative, scatter plots provide important visual context by allowing readers to:
 - a. Evaluate whether relationships are linear or nonlinear (in particular, the current study only considered linear correlations)
 - b. Detect outliers that may influence the correlation
 - c. Identify potential subgroups or clusters (e.g., based on age or sex)
 - d. Observe patterns of heteroscedasticity (variability in the spread of data)

Including such plots would enhance the interpretability and transparency of the reported associations.

Reviewer #2

(Remarks to the Author)

The authors present CSF biomarker data from 1692 older individuals (760 cognitively normal; 932 impaired) enrolled in ADNI or WRAP that supports interpretation / reconceptualization of total tau as a biomarker of synaptic dysfunction not neurodegeneration. This assertion is supported by converging lines of evidence demonstrating 1) a stronger correlation between CSF t-tau and markers of synaptic degeneration (neurogranin and SNAP-25) vs markers of neurodegeneration (NfL); 2) that most variance in t-tau levels is accounted for by changes in Ng and SNAP-25 levels; and, 3) T-tau levels were highest in groups with synaptic degeneration (S+), regardless of measures of neurodegeneration (N +/-). The authors assert that these findings support use of CSF t-tau to measure “early synaptic degeneration rather than late axonal/neuronal

degeneration.” These findings are interesting, although the implications of these findings are understated. It will be important for the Authors to clarify how they envision a marker of “early synaptic degeneration” could be applied in clinical and research settings.

I raise the following additional points for consideration:

1. Given the assertion that t-tau may reflect early synaptic degeneration, it would be interesting to know if the relationships reported in this manuscript varied by global CDR stage (CDR 0.5 vs ≥ 1) or clinical diagnosis (i.e., MCI vs AD dementia)?
2. Did the Authors consider exploring the association between CSF t-tau and other conventional measures of early synaptic dysfunction / degeneration in AD—namely FDG-PET AD-ROI for the subset of ADNI participants with these data?
3. Broader discussion is encouraged to explore differences between findings in this study and others.
 - Mielke et al 2021 (Alz & Dement). Elevated T-tau but not Ng was associated with decreasing temporal lobe and AD meta-ROI cortical thickness in participants in the Mayo Clinic Study of Aging.
 - Dhiman et al 2020 (Alz & Dement: DADM). Reasonably high correlations between CSF t-tau and NfL.
4. It would also be useful to discuss what this study adds to the literature compared to other published works that report similarly high correlations between CSF t-tau and markers of synaptic dysfunction (namely Ng).
 - Vidal-Pineiro et al. 2022 (Neurobiology of Aging). High correlation between CFS t-tau and Ng (0.83).
 - Mattsson et al. 2016 (EMBO Mol Med). High correlation between CFS t-tau and Ng (0.795) in smaller sample of ADNI cohort.
5. The Authors reference their relatively diverse WRAP cohort in the last paragraph of the Discussion, but do not present any data in the Results to support this statement. Additional participant details should be added to Table 1 to support these points and inform generalization of results.

Minor points for further characterization:

- Figure 3 (bar graph) is unnecessary. Numbers could be directly reported in the text and the figure removed.
- Labels in Fig 4 are incorrect. N+/S+ is presented twice. I believe the first one should be N+/S-.
- Repetition across manuscript should be reduced (e.g., cohort size reported in introduction, methods, and results—could be limited to results; key findings repeated in paragraphs 1 and 2 of Discussion).

Reviewer #3

(Remarks to the Author)

The objective of this study was to show that CSF total tau can be used as a proxy of synaptic degeneration. The study uses data from two large ($n=1692$), well-characterised cohorts. The novelty lies in the approach the authors have taken in analyzing these pre-existing data. For many years, CSF total tau (t-tau) has been conceptualised as a tau-mediated neurodegeneration biomarker. Here the authors argue that CSF t-tau was more closely associated with classical biomarkers of synaptic degeneration than those of neuronal degeneration and that its variability was largely explained by synaptic biomarkers. They go on to argue that CSF t-tau may not be a good neurodegeneration proxy (as is common in observational and interventional AD studies) but rather, that it may be a good proxy of synaptic degeneration that precedes neurodegeneration early in AD. The study is well-powered and the data reported replicate well across both cohorts. This is an interesting premise that is largely supported by the data provided but there are some issues with the manuscript (errors, missing explanations, clarity on statistical approaches) that make the study difficult to assess in its current form.

Major issues:

Regarding Biomarker classification:

P6. Line 130. “Participants were classified as A β -positive using CSF A β 1-42 below 976.6 pg/ml (ADNI)³⁸ and CSF A β 42/A β 40 below 0.046 (WRAP).³⁷” The authors should revise the references cited here as they do not seem to provide the cutoffs for those cohorts. If new cut-offs were calculated, this should be stated and the sensitivity, specificity and accuracy for the cutoff should be provided. Moreover, I recommend that the authors use the same biomarker for amyloid positivity in both cohorts or provide an explanation of why CSF A β 1-42 was used for ADNI but CSF A β 42/A β 40 was used for WRAP, when A β 42/A β 40 data is available for ADNI. Using different biomarkers for amyloid positivity for the 2 cohorts, which are then merged for analysis is not appropriate.

Regarding biomarkers selected for the study:

Why was A β 42 used in the correlation analyses when A β 42/40 is the generally accepted biofluid marker for diagnosis, staging and prognosis of AD (see Jack et al *Alzheimers Dement.* 2024 Aug;20(8):5143-5169)?

A comment on whether these associations hold for phosphorylated tau (ptau181) with synaptic markers are lacking. This would be interesting since ptau181 is more commonly recognised as a pathological marker of tau tangles whereas t-tau may reflect tau secretion related to neuronal damage or secretion. As treatments targeting tau pathology are being trialled, a biomarker of synapse degeneration (be it Ng or t-tau) that is not itself a proxy of the pathology being targeted would be very useful. Although I expect the results to be similar (t-tau and ptau181 often correlate very well), a comment on this in the discussion, could be helpful.

The authors use hippocampal volume and NfL as neurodegeneration markers in this study. When classifying participants as

N+/N-, the authors used HC volume and not NfL. HC volume reflects the accumulated atrophy of a specific AD-related region up to the point where measured whereas NfL is an active measure of axonal degeneration across the CNS at the timepoint measured. Thus, in contrast to the rest of the paper, for the purposes of the final analyses, the authors are defining neurodegeneration as only occurring in the hippocampus. There may be a floor effect in HC loss, and HC volume does not capture neuronal loss in other brain regions. This is not tackled in the manuscript.

Regarding statistical analyses:

The distribution of the data is not commented on nor how suitable the data were for parametric analyses. Furthermore, in the Statistical Analyses section, it states that Pearson's correlation test was used for correlations. However, both Figure 1 and Supplementary Figure 1 legends state that Spearman correlations were used. Supplementary Table 3 states that Pearson was used. Which was it and what was the justification for the chosen method? They are not equivalent.

Regarding figures:

In the legend to Figure 1: "All variables were log-transformed and z-scored, and HCV and CSF A β 1-42 were further inverted (*-1)." What does (*-1) mean? Why were HCV and CSF A β 1-42 values inverted? The correlation coefficients are negative for A β 1-42 (as you would expect) suggesting they were not inverted. The legend needs revising for clarity. Same for Supplementary Figure 1 legend.

There are errors in Figure 4. and supplementary Figure 3. The last two categories are both labelled as N+/S+.

Minor issues.

Information is lacking in the Study population section relating to which ADNI phase was used and download date.

Participants were considered S+ based on either Ng or SNAP25 values above median from CU A β -negative individuals.⁵⁰

I do not think ref 50 is correct here.

I have spotted several references that do not seem to be correct (eg., 33, 34, 50?). This should be revised.

P8. Line 182. "Overall, we observed weak correlations between neurodegeneration synaptic biomarkers." Should read "Overall, we observed weak correlations between neurodegeneration and synaptic biomarkers"

In the discussion Page 2 line 254. There is repetition of previous paragraph.

Version 1:

Reviewer comments:

Reviewer #1

(Remarks to the Author)

The authors have addressed my concerns. I do not have further comments.

Reviewer #2

(Remarks to the Author)

The Authors have added additional data and completed further analyses yielding an improved manuscript that reasonably supports the conclusions. I have no further comments or recommendations for revision.

Reviewer #3

(Remarks to the Author)

The changes/clarifications made in the revised manuscript and letter to reviewers have addressed all my concerns.

Reviewer #1 (Remarks to the Author):

This manuscript by Soares et al., studied associations between CSF t-tau and synaptic degeneration (i.e., Ng and SANP25), as well as neurodegeneration markers (i.e., hippocampal volume and NfL) in 760 cognitively unimpaired and 932 cognitively impaired participants. Data from two independent cohorts were used to demonstrate that CSF t-tau showed higher linear correlations with synaptic degeneration markers (quantified by CSF Ng and SANP25) than with neurodegeneration markers (quantified by hippocampal volume and NfL). This is an interesting paper using reasonable approaches to address timely questions in the AD field, which largely indicates experience and competence among the authors. Overall, this study is well-designed, and the manuscript is well written. I only have a few minor concerns on the methods and results interpretation, which could be addressed in a revision.

Response: Thank you for your thoughtful comments and suggestions. We have made the following revisions in response to your feedback: 1) sex stratification analysis; 2) correction for multiple comparisons; 3) clarification of group definitions; 4) visualization of correlation results. We believe these revisions have strengthened the manuscript, and we appreciate your constructive feedback. See below for more details:

Comment 1. Although the CSF measures were orthogonalized with respect to age and sex, it is still possible that sex-specific multivariate effects may influence the associations. To address this, could the authors consider reanalyzing the associations stratified by sex (i.e., in females and males separately)?

Response: Thank you for your valuable suggestion. We re-estimated every primary model stratified by sex (age-, cohort-adjusted) and now report the full outputs in the results (page 6 line 7 and page 7 line 14) with Supplementary Figure 3, Supplementary Tables 8 and 14. In CU participants, both men and women showed the same qualitative hierarchy—CSF t-tau associated most strongly with the synaptic markers Ng ($\beta=2.495$ and 2.521 in males) and SNAP25 ($\beta=2.474$ in females, 1.921 in males), whereas its links to neurodegeneration biomarkers were smaller (NfL: $\beta=2.262$ vs 1.631) or non-significant (HCV). Notably, in CU individuals the association with SNAP25 was higher in females than in males. In CI individuals, the pattern persisted but a female-specific amplification emerged for the CSF t-tau–Ng association ($\beta = 3.37$, vs 2.48 in males); HCV ($\beta=0.826$ in females, 0.894 in males), NfL ($\beta=1.407$ in females, 1.318 in males), SNAP25 ($\beta=2.275$ in females, 1.895 in males). Thus, stratification confirms our core finding—that CSF t-tau preferentially tracks synaptic degeneration in both sexes—and indicates only a modest, stage-specific sex effect for Ng, which we have now highlighted in the Discussion (page 10 line 111)

“Furthermore, we found that females exhibit a stronger association of CSF t-tau with SNAP25 in the CU group, and with Ng in the CI group. Although one study found no sex differences in SNAP25 levels in A β -positive CU individuals, data on this biomarker remain limited.²⁵ Alternatively, different groups have reported increased Ng in females, both in combined CU and CI samples regardless of A β status,¹⁵ and in a sample stratified by A β status,²⁶ and specifically in CU A β -positive individuals.²⁵ These findings indicate that CSF t-tau is more strongly linked to synaptic biomarkers in females, starting from the early stages of AD.”

β estimates with 95% confidence intervals from linear regressions, adjusted for age, cohort and sex where applicable, showing the association of CSF t-tau with biomarkers of established neurodegeneration (HCV, CSF NfL) and synaptic dysfunction (CSF Ng and SNAP25) stratified by cognitive status and sex (a) or by clinical diagnosis (b), across cohorts. SNAP25 was not modeled in the dementia group due to limited sample size. Cognitively unimpaired (CU). Cognitively impaired (CI). Hippocampal volume (HCV). Neurogranin (Ng). Neurofilament light chain protein (NfL).

Comment 2. Were the reported P-values corrected for multiple comparisons using methods such as False Discovery Rate (FDR) or Bonferroni correction? If not, it would be important to acknowledge this as a limitation in the manuscript.

Response: Thank you for pointing this limitation. We have now applied a Bonferroni adjustment across all pair-wise tests ($\alpha = 0.05$) and re-computed the p-values for every correlation estimate. Importantly, the pattern of findings is unchanged: all effects highlighted in the original manuscript remain significant after correction, while previously non-significant associations remain so. The adjusted p-values (P_{adj}) are included in the supplementary tables and legends, and the results have been updated in the main text, as well as the Methods.

Comment 3. In the Statistical Analysis section, please specify which “four groups” were used to compare mean levels of t-tau. While this becomes clear in Figure 4, it should be explicitly described in the Methods section for clarity.

Response: Thank you for your comment. We have clarified the groups used for comparing mean levels of CSF t-tau in the Statistical Analysis section. The following description has been added to the Methods (page 15 line 14) section for greater clarity:

“Additionally, within each cohort, four groups were generated to compare mean levels of t-tau according to their status of synaptic degeneration (S) and neurodegeneration (N): N⁻/S⁻, N⁻/S⁺, N⁺/S⁻ and N⁺/S⁺.”

Comment 4. Could the authors consider visualizing the correlation results using scatter plots? While correlation coefficients (e.g., $r = 0.8$) are informative, scatter plots provide important visual context by allowing readers to:

a. Evaluate whether relationships are linear or nonlinear (in particular, the current study only

considered linear correlations)

b. Detect outliers that may influence the correlation

c. Identify potential subgroups or clusters (e.g., based on age or sex)

d. Observe patterns of heteroscedasticity (variability in the spread of data)

Including such plots would enhance the interpretability and transparency of the reported associations.

Response: Thank you for this helpful suggestion. We agree that scatter plots provide valuable visual context. However, given the large number of pairwise correlations among seven biomarkers across both CU and CI groups, including scatter plots for every combination would result in an overwhelming number of plots (>80). To balance completeness and readability, we have previously included a comprehensive table in the Supplementary Tables 3-5 reporting all correlation coefficients with means and 95% confidence intervals. We recognize that this format does not visualize individual data points, but it provides a clear and concise summary of all relationships. Nevertheless, to address your comment and highlight the key associations central to our paper, we have now added a new Supplementary Figure 2 presenting scatter plots of CSF t-tau versus HCV, NfL, Ng, and SNAP25 in CU and CI across cohorts. We believe this addition improves interpretability while maintaining a manageable figure set.

Associations of CSF t-tau with biomarkers of neuronal and synaptic degeneration

a Cognitively Unimpaired

b Cognitively Impaired

Scatter plots from linear regressions adjusted for age and sex, showing the association of CSF t-tau with biomarkers of established neurodegeneration (HCV, CSF NfL) and synaptic dysfunction (CSF Ng and SNAP25) in (a) CU and (b) CI individuals across two independent cohorts. Cognitively unimpaired (CU). Cognitively impaired (CI). Hippocampal volume (HCV). Neurogranin (Ng). Neurofilament light chain protein (NfL).

Reviewer #2 (Remarks to the Author):

The authors present CSF biomarker data from 1692 older individuals (760 cognitively normal; 932 impaired) enrolled in ADNI or WRAP that supports interpretation / reconceptualization of total tau as a biomarker of synaptic dysfunction not neurodegeneration. This assertion is supported by converging lines of evidence demonstrating 1) a stronger correlation between CSF t-tau and markers of synaptic degeneration (neurogranin and SNAP-25) vs markers of neurodegeneration (NfL); 2) that most variance in t-tau levels is accounted for by changes in Ng and SNAP-25 levels; and, 3) T-tau levels were highest in groups with synaptic degeneration (S+), regardless of measures of neurodegeneration (N +/-). The authors assert that these findings support use of CSF t-tau to measure “early synaptic degeneration rather than late axonal/neuronal degeneration.” These findings are interesting, although the implications of these findings are understated. It will be important for the Authors to clarify how they envision a marker of “early synaptic degeneration” could be applied in clinical and research settings. I raise the following additional points for consideration:

Response: Thank you for your thoughtful comments and suggestions. We have made the following revisions in response to your feedback: 1) CDR/clinical stratification analysis; 2) FDG-PET AD-ROI analysis; 3-4) broader discussion; 5) WRAP cohort diversity; 6) Figure 3; 7) Labels in Figure 4; 8) reduction of repetition across the manuscript. We believe these revisions have strengthened the manuscript, and we appreciate your constructive feedback. See below for more details:

Comment 1. Given the assertion that t-tau may reflect early synaptic degeneration, it would be interesting to know if the relationships reported in this manuscript varied by global CDR stage (CDR 0.5 vs ≥ 1) or clinical diagnosis (i.e., MCI vs AD dementia)?

Response: We thank the reviewer for this insightful comment. We re-estimated the age- and sex-adjusted models separately in the two impaired groups—MCI or CDR = 0.5 and AD dementia or CDR > 0.5—and found that the relationships do not vary meaningfully by CDR stage or clinical diagnosis reported in the results (page 7 line 16): in both subgroups CSF t-tau remains most strongly linked to the synaptic markers (Ng > SNAP25), shows a mid-range association with NfL, and displays only a weak association with HCV, exactly the hierarchy observed in the pooled CI sample. Effect sizes are numerically higher in dementia, as expected with advancing pathology, but the rank order is unchanged; the only limitation is that SNAP25 could not be modelled in dementia because fewer than 10 cases had that measure. These stratified results—now provided in Supplementary Figure 3 and Supplementary Table 15—confirm that the core pattern holds across CDR stages and clinical diagnoses.

β estimates with 95% confidence intervals from linear regressions, adjusted for age, cohort and sex where applicable, showing the association of CSF t-tau with biomarkers of established neurodegeneration (HCV, CSF NfL) and synaptic dysfunction (CSF Ng and SNAP25) stratified by cognitive status and sex (**a**) or by clinical diagnosis (**b**), across cohorts. SNAP25 was not modeled in the dementia group due to limited sample size. Cognitively unimpaired (CU). Cognitively impaired (CI). Hippocampal volume (HCV). Neurogranin (Ng). Neurofilament light chain protein (NfL).

Comment 2. Did the Authors consider exploring the association between CSF t-tau and other conventional measures of early synaptic dysfunction / degeneration in AD—namely FDG-PET AD-ROI for the subset of ADNI participants with these data?

Response: Thank you for suggesting incorporating FDG-PET. We ran additional age-, sex-adjusted linear-regression models in the ADNI subset with FDG scans and CSF t-tau. The full results (page 6 line 5 and page 7 line 14) are now presented in Supplementary Table 13. Key findings are:

CU participants: FDG-PET showed no significant association with CSF t-tau ($p = 0.20$) or any other biomarker (HCV, CSF NfL, Ng, SNAP25). CI participants: FDG-PET was significantly related to CSF t-tau ($\beta = -0.24, p < 0.001$), yet this effect was ~50 % smaller than its association with hippocampal volume ($\beta = -0.46, p < 0.001$) and explained far less variance ($R^2 = 0.13$ vs. 0.29). No relationship emerged with Ng or SNAP25, although the sample size is considerably smaller ($p > 0.05$; $n = 23$ for SNAP25 model).

These results reinforce our central conclusion: CSF t-tau aligns more closely with synaptic proteins than with structural neurodegeneration markers. FDG-PET’s mixed biological signal explains why its association with CSF t-tau falls between those two domains and why adding it to the primary analyses would blur, rather than clarify, the mechanistic distinction we draw.

Comment 3. Broader discussion is encouraged to explore differences between findings in this study and others.

- Mielke et al 2021 (Alz & Dement). Elevated T-tau but not Ng was associated with decreasing temporal lobe and AD meta-ROI cortical thickness in participants in the Mayo Clinic Study of Aging.

- Dhiman et al 2020 (Alz & Dement: DADM). Reasonably high correlations between CSF t-tau and NfL.

Response: Thank you for the valuable suggestion. We initially focused on studies more directly comparable to our analysis, specifically those examining CU or CI individuals separately, which was not the case with (Dhiman et al., 2020). Even though Dhiman et al presents a challenge in comparing results, as it combined both CU and CI participants in their analysis, we added it to the discussion together with (Mielke et al., 2021) study, which was already referenced in the manuscript, but now have been highlighted in the discussion (page 9 line 11) where it reads:

“In CI individuals, we observed a modest association between CSF t-tau and HCV, CSF NfL and FDG-PET, while in CU individuals, CSF t-tau was only moderately associated with CSF NfL. These results align with prior studies in CI individuals which showed also low-modest associations with measures of FDG-PET,(Haque et al., 2023) HCV and cortical thickness(Haque et al., 2023; Mielke et al., 2021; Pereira et al., 2017) and plasma NfL,(Pereira et al., 2017) but differ from studies that found significant associations in the CU population(Ebenau et al., 2022) or observed different degrees of correlation in a CU and CI combined sample.(Dhiman et al., 2020; Illán-Gala et al., 2018)”

Comment 4. It would also be useful to discuss what this study adds to the literature compared to other published works that report similarly high correlations between CSF t-tau and markers of synaptic dysfunction(namely Ng).

- Vidal-Pineiro et al. 2022 (Neurobiology of Aging). High correlation between CFS t-tau and Ng (0.83).

- Mattsson et al. 2016 (EMBO Mol Med). High correlation between CFS t-tau and Ng (0.795) in smaller sample of ADNI cohort.

Response: We added the novel contribution of our work in the discussion (page 10 line 2) where it reads: “Therefore, our study not only reaffirms the strong link between CSF t-tau and synaptic biomarkers but also indicates that CSF t-tau is more closely associated with synaptic degeneration than neurodegeneration, offering new interpretation to its abnormality in AD progression.”

We also included Mattsson (16) et al and Vidal-Pineiro (28) to the discussion (page 9 line 9) where it reads:

“We found strong associations between CSF t-tau with CSF Ng and SNAP25 in both CU and CI individuals, reinforcing mounting evidence showing a strong link between CSF t-tau and Ng,^{16,17,25–28} SNAP25,^{25,26} GAP43,^{25,26} synaptotagmin-1,²⁶ and neuropentraxin-2.^{25”}

Comment 5. The Authors reference their relatively diverse WRAP cohort in the last paragraph of the Discussion, but do not present any data in the Results to support this statement. Additional participant details should be added to Table 1 to support these points and inform generalization of results.

Response: Thank you for this helpful comment. We have now updated Table 1 and relevant Supplementary Tables to include both race and ethnicity information for the ADNI and WRAP cohorts, ensuring consistent reporting across cohorts. In doing so, we also identified and corrected an important underestimation of the

number of White, non-Hispanic participants, and have revised the diversity statement in the Discussion (page 12 line 5) accordingly:

“Additionally, the WRAP cohort is mainly composed of CU and MCI individuals, and like the currently available ADNI cohort, is also predominantly composed of White participants. It would be desirable to replicate these results in a more diverse general population.”

While WRAP is less racially diverse than we originally stated, the updated tables now provide transparent data to support interpretation and generalizability of findings. We appreciate the reviewer’s suggestion to clarify this.

Minor points for further characterization:

Comment 6 - Figure 3 (bar graph) is unnecessary. Numbers could be directly reported in the text and the figure removed.

Response: Thank you for the suggestion. While we agree that the numerical values could be conveyed in the text alone, we have chosen to retain Figure 3 because it provides an immediate and intuitive visual comparison of the relative contributions of each biomarker to t-tau variance. This visual representation complements the statistical results and helps emphasize the differential strength of associations, which is central to the study’s interpretation. We believe the figure adds clarity for readers and improves accessibility of the findings.

Comment 7 - Labels in Fig 4 are incorrect. N+/S+ is presented twice. I believe the first one should be N+/S-.

Response: We appreciate the attention to detail in our manuscript. The labels were corrected in Figure 4 and Supplemental Figure 6.

Figure 4. CSF t-tau is increased in individuals with abnormal synaptic degeneration regardless of concomitant presence of neurodegeneration. (a) Bar graphs show the distribution of CU, MCI and dementia across synaptic

degeneration (S) and neurodegeneration (N) groups. N positivity was based on HCV and S positivity was based on CSF Ng (N⁻S⁻: n = 85; N⁻S⁺: n = 137; N⁺S⁻: n = 110; S⁺N⁺: n = 252). Cutoffs were calculated as above or below median from CU A β individuals. (b) Violin plots show CSF t-tau levels in individuals with abnormal S and/or N in the whole population (CU and CI) across all cohorts. Linear regression model with dummy variables, adjusted for age, sex, cognitive status, A β burden and cohort was employed to compare CSF t-tau levels across groups. * P-value < 0.05. Amyloid- β (A β). Cognitively unimpaired (CU). Hippocampal volume (HCV). Mild cognitive impairment (MCI). Neurodegeneration (N). Neurogranin (Ng). Synaptic degeneration (S).

Comment 8 - Repetition across manuscript should be reduced (e.g., cohort size reported in introduction, methods, and results—could be limited to results; key findings repeated in paragraphs 1 and 2 of Discussion).

Response: Thank you for this helpful suggestion. We have removed redundant mentions of cohort size from the Introduction and Methods, limiting this information to the Results section. In addition, we revised the Discussion to reduce repetition of key findings between the first and second paragraphs, streamlining the text for clarity and conciseness (page 9 line 2):

“In this study, we investigated the association of CSF t-tau with markers of synaptic and axonal/neuronal degeneration across the aging and the AD spectrum in two independent large cohorts. Our findings suggest that CSF t-tau was more closely associated with classical biomarkers of synaptic degeneration than those of neuronal degeneration. Notably, we found evidence that abnormalities in biomarkers of synaptic degeneration, even in the absence of neuronal degeneration abnormality indexed by HCV, were strongly associated with increased CSF t-tau levels. Thus, CSF t-tau shows the potential to be used as a proxy of synaptic degeneration.

We found strong associations between CSF t-tau with CSF Ng and SNAP25 in both CU and CI individuals, reinforcing mounting evidence showing a strong link between CSF t-tau and Ng^{15,16,24–27} SNAP25,^{24,25} GAP43,^{24,25} synaptotagmin-1,²⁵ and neuropentraxin-2.²⁴ In CI individuals, we observed a modest association between CSF t-tau and HCV, CSF NfL and FDG-PET, while in CU individuals, CSF t-tau was only moderately associated with CSF NfL. These results align with prior studies in CI individuals showing low-moderate associations with FDG-PET,²⁸ HCV and cortical thickness^{28–30} and plasma NfL,³⁰ but differ from reports that found significant associations in the CU population³¹ or observed stronger correlations in a CU and CI combined sample.^{32,33} Given the limited evidence in CU populations and variability across studies, further research is needed to clarify t-tau’s role in preclinical stages. Altogether, these findings suggest that CSF t-tau is well associated with synaptic biomarkers before cognitive impairment, but only moderately - and most frequently only in symptomatic individuals - with measures of atrophy, neuronal injury and glucose uptake.”

Reviewer #3 (Remarks to the Author):

The objective of this study was to show that CSF total tau can be used as a proxy of synaptic degeneration. The study uses data from two large (n=1692), well-characterised cohorts. The novelty lies in the approach the authors have taken in analyzing these pre-existing data. For many years, CSF total tau (t-tau) has been conceptualised as a tau-mediated neurodegeneration biomarker. Here the authors argue that CSF t-tau was more closely associated with classical biomarkers of synaptic degeneration than those of neuronal degeneration and that its variability was largely explained by synaptic biomarkers. They go on to argue that CSF t-tau may not be a good neurodegeneration proxy (as is common in observational and interventional AD studies) but rather, that it may be a good proxy of synaptic degeneration that precedes neurodegeneration early in AD. The study is well-powered and the data reported replicate well across both cohorts. This is an interesting premise that is largely supported by the data provided but there are some issues with the manuscript (errors, missing explanations, clarity on statistical approaches) that make the study difficult to assess in its current form.

Response: We thank the reviewer for their constructive feedback. In response, we have made the following revisions: 1) corrected the reference for the A β cutoff; 2) explained the use of different A β biomarkers across cohorts and the rationale for using A β 42 in correlations; 3) added analyses including p-tau181 and plasma p-tau217; 4) addressed the use of HCV only for N+ classification by adding NfL-based analyses; 5) clarified model assumptions and justified the use of Pearson correlations, correcting figure legend discrepancies; 6-7) clarified the meaning and rationale for biomarker inversion in figures; 8-9) notation and biomarker inversion clarification; and 10-13) resolved minor issues related to labeling errors, ADNI phase info, references, wording; and 14) redundancy in the discussion.

Major issues:

Regarding Biomarker classification:

Comment 1. P6. Line 130. “Participants were classified as A β -positive using CSF A β 1-42 below 976.6 pg/ml (ADNI)38 and CSF A β 42/A β 40 below 0.046 (WRAP).37” The authors should revise the references cited here as they do not seem to provide the cutoffs for those cohorts. If new cut-offs were calculated, this should be stated and the sensitivity, specificity and accuracy for the cutoff should be provided.

Response: Thank you for your important observation. Both cutoffs for WRAP and ADNI correspond to previously published values. The reference cited for the WRAP cohort is correct (Van Hulle, C. et al. An examination of a novel multipanel of CSF biomarkers in the Alzheimer’s disease clinical and pathological continuum. *Alzheimer’s & Dementia* 17, 431–445 (2021). The reference cited for ADNI CSF A β 42 cutoff has been updated to the appropriate source where it can be found in Supplementary table 8 of the reference:

Hansson, O. et al. CSF biomarkers of Alzheimer’s disease concord with amyloid- β PET and predict clinical progression: A study of fully automated immunoassays in BioFINDER and ADNI cohorts. *Alzheimer’s & Dementia* 14, 1470–1481 (2018).

This publication provides the validated cutoff of 976.6 pg/mL for A β 42 in the ADNI cohort, based on concordance with amyloid PET.

Comment 2. Moreover, I recommend that the authors use the same biomarker for amyloid positivity in both cohorts or provide an explanation of why CSF A β 1-42 was used for ADNI but CSF A β 42/A β 40 was used for WRAP, when A β 42/A β 40 data is available for ADNI. Using different biomarkers for amyloid positivity for the 2 cohorts, which are then merged for analysis is not appropriate.

Response: We appreciate the concern and agree that, ideally, the same amyloid-positivity marker would be applied across cohorts. However, we would like to clarify a few aspects. First, we used previously published cutoffs for ADNI (Hansson et al., 2018) and WRAP (Van Hulle et al., 2021) cohorts. Secondly, other publications have used different methods for amyloid positivity (Ashton et al., 2021; Bellaver et al., 2023). Moreover, even though there are published cutoffs for CSF A β 42/A β 40, in our study, we prioritized individuals with measures of synaptic degeneration (SNAP25 and Ng), which limited the sample size from ADNI. The ADNI core biomarker data were obtained from the UPENNBIOMK_ROCHE_ELECSYS.csv file. While both A β 42 and A β 40 are available in ADNI, approximately 70% of participants with synaptic markers lack A β 40 data, whereas A β 42 is available for all. To maximize statistical power, we therefore used the validated A β 42 alone in ADNI rather than the ratio. Thus, ADNI's 1 408 participants selected in this study have CSF A β 42 while only 434 (\approx 30 %) have the additional A β 40 measurement needed for the A β 42/40 ratio, so adopting the ratio would have reduced the ADNI sample—and therefore power for our synaptic analyses. Importantly, we performed additional analysis within the 434 individuals who possess both A β biomarkers, and confirmed that the ratio of A β -positivity did not materially change: 47.5 % were A β -positive using the published A β 42 cutoff (\leq 976.6 pg/ml) versus 43.5 % with the ratio cutoff ($<$ 0.057, (Leuzy et al., 2023), and McNemar's test indicated no significant discordance between the two proportions ($\chi^2 = 3.61$, $p = 0.058$). In contrast, WRAP has a published threshold based on the A β 42/40 ratio and no cases without standalone A β 42. Therefore, using the established ratio maximized WRAP's sample size and diagnostic validity. Although the ratio shows higher concordance with amyloid-PET in general (e.g., Hansson et al., 2018), for the present aims we prioritized sample size and comparable positivity rates: each cohort retained its most complete and cohort-validated marker, and these cohort-specific definitions were used only to derive the binary CU A– reference groups. Different continuous measures of CSF A β , namely CSF A β 42/40 ratio and CSF A β 42, were not merged or entered together in any analytic model (e.g. the correlation matrix or as a covariate) —thus preserving both statistical power in ADNI and methodological consistency within WRAP while avoiding cross-assay conflation.

Finally, to address this disparity in the manuscript, we included a comment in the methods (page 14 line 9):

Methods: “In WRAP, we used the CSF A β 42/A β 40 ratio to leverage published cutoff.⁴⁸ We used CSF A β 42 alone in ADNI to preserve sample size due to limited availability of concomitant synaptic biomarkers and A β 40 data in the table “UPENNBIOMK_ROCHE_ELECSYS.csv”.

Regarding biomarkers selected for the study:

Comment 3. Why was A β 42 used in the correlation analyses when A β 42/40 is the generally accepted biofluid marker for diagnosis, staging and prognosis of AD (see Jack et al *Alzheimers Dement.* 2024 Aug;20(8):5143-5169)?

Response: Thank you for your suggestion. As noted previously, we used CSF A β 42 in ADNI and CSF A β 42/A β 40 in WRAP due to cohort-specific data availability. While ADNI had A β 42 available for all participants, only a subset had both A β 42 and A β 40 measurements, and using only A β 42 preserved statistical power for our analyses. Conversely, WRAP had A β 42/40 ratio data for all participants and using this maximized sample size and diagnostic validity. When combining cohorts, we applied cohort-specific definitions of amyloid positivity solely to define the CU A- control group. Importantly, we did not merge or jointly analyze A β 42 and A β 42/40 ratio values in any statistical models (e.g., correlation matrices or covariates), thereby preserving internal consistency and allowing for appropriate cohort-specific analyses.

Comment 4. A comment on whether these associations hold for phosphorylated tau (ptau181) with synaptic markers are lacking. This would be interesting since ptau181 is more commonly recognised as a pathological marker of tau tangles whereas t-tau may reflect tau secretion related to neuronal damage or secretion. As treatments targeting tau pathology are being trialled, a biomarker of synapse degeneration (be it Ng or t-tau) that is not itself a proxy of the pathology being targeted would be very useful. Although I expect the results to be similar (t-tau and ptau181 often correlate very well), a comment on this in the discussion, could be helpful.

Response: Thank you for raising this important point. We have now tested p-tau181 as well as plasma p-tau217 alongside CSF t-tau in fully adjusted linear models of each synaptic marker (Ng and SNAP-25) stratified by clinical stage (CU and CI) in Supplementary Figure 4 and Supplementary Table 9. Accordingly, we report significant associations for p-tau181, but no significant associations with plasma p-tau217 and we added a comment in the discussion as reads:

“Moreover, consistent with studies showing a strong association between CSF t-tau and CSF p-tau181,^{24,27,29} we also found significant associations between synaptic biomarkers and CSF p-tau181 but not with plasma p-tau217. This suggests future studies could explore the role of synaptic degeneration in p-tau181 signal. The fact that plasma p-tau217 was only available with concomitant synaptic biomarkers in the WRAP with composed primarily of CU may be playing a role in this lack of association.”

Comment 5. The authors use hippocampal volume and NfL as neurodegeneration markers in this study. When classifying participants as N+/N-, the authors used HC volume and not NfL. HC volume reflects the accumulated atrophy of a specific AD-related region up to the point where measured whereas NfL is an active measure of axonal degeneration across the CNS at the timepoint measured. Thus, in contrast to the rest of the paper, for the puposes of the final analyses, the authors are defining neurodegeneration as only occurring in the hippocampus. There may be a floor effect in HC loss, and HC volume does not capture neuronal loss in other brain regions. This is not tackled in the manuscript.

Response: Thank you very much for this important and insightful comment regarding our choice of neurodegeneration markers. We agree with the interpretation that HCV and CSF NfL may be capturing different aspects of neurodegeneration: HCV reflects accumulated, region-specific atrophy over time (Llibre-Guerra et al., 2019) whereas NfL represents ongoing, active axonal injury across the CNS (Zetterberg et al., 2016).

We have now performed an additional analysis substituting NfL for HCV, presented in the results (page 8 line 18) supported by Supplementary Figure 6 and Supplementary Table 20-21 and added to the discussion (page line 10 line 24). These results closely mirror the primary findings: synaptic abnormalities remain more strongly associated with elevated CSF t-tau levels compared to neurodegeneration, with the exception that the presence of both is associated with even higher levels of CSF t-tau.

a Distribution of diagnosis across N/S groups

b CSF t-tau levels across N/S groups

(a) Bar graphs show the distribution of CU, MCI and dementia across synaptic degeneration (S) and neurodegeneration (N) groups. N positivity was based on HCV or CSF NfL and S positivity was based on CSF Ng (n = 584) or SNAP25 (n = 280). Cutoffs were calculated as above or below median from CU A β ⁻ individuals. (b) Violin plots show CSF t-tau levels in individuals with abnormal S and/or N in the whole population (CU and CI) across all cohorts. Linear regression model with dummy variables, adjusted for age, sex, cognitive status, amyloid burden and cohort was employed to compare CSF t-tau levels across groups. *P-value < 0.05. Amyloid- β (A β). Cognitively unimpaired (CU). Hippocampal volume (HCV). Mild cognitive impairment (MCI). Neurodegeneration (N). Neurofilament light protein (NfL). Neurogranin (Ng). Synaptic degeneration (S).

Discussion now reads (page line 10 line 24):

“The observation that CSF NfL abnormalities alone were associated with elevated CSF t-tau — albeit less strongly than when accompanied by synaptic abnormalities — suggests that CSF NfL may serve as an earlier neurodegeneration biomarker, potentially overlapping with synaptic dysfunction. This adds to the growing evidence base documenting inconsistencies among neurodegeneration markers.^{31,33”}

Regarding statistical analyses:

Comment 6. The distribution of the data is not commented on nor how suitable the data were for parametric analyses.

Response: Thank you for this important point. Given our large sample sizes across cohorts and clinical groups, the central limit theorem supports the use of parametric linear models, and the estimates can be considered robust and approximately normally distributed. Additionally, we conducted diagnostic checks on model residuals—including visual inspection of Q-Q plots and tests for homoscedasticity—and confirmed that assumptions of normality and equal variance were reasonably met for all primary analyses. We have added a brief statement to the Methods section to clarify this (page 16 line 2):

“Assumptions were validated through standard diagnostic checks, including histograms of the residuals, Q-Q plots, and scatterplots of residuals versus fitted values. These checks confirmed that the assumptions were met and deemed reasonable.”

Comment 7. Furthermore, in the Statistical Analyses section, it states that Pearson’s correlation test was used for correlations. However, both Figure 1 and Supplementary Figure 1 legends state that Spearman correlations were used. Supplementary Table 3 states that Pearson was used. Which was it and what was the justification for the chosen method? They are not equivalent.

Response: We appreciate the reviewer for noticing the discrepancies. Pearson’s correlation test was used for analyses, in Figure 1, Supplementary Figure 1, and Supplementary Table 3-5. The figure legends have been updated for consistency.

Pearson’s method was selected given the assumption of linearity and the sufficient sample size in the CI group ($n > 90$), particularly for the SNAP25 and HCV correlations. This approach is appropriate when linear relationships are expected and provides greater statistical power under these conditions. The rationale has been clarified in the Statistical Analyses (page 15 line 2) section as mentioned in the previous comment.

Regarding figures:

Comment 8. In the legend to Figure 1: “All variables were log-transformed and z-scored, and HCV and CSF A β 1-42 were further inverted (*-1).” What does (*-1) mean?

Response: Thank you for your comment. The notation “(*-1)” was intended to indicate that values were multiplied by -1 to invert their direction, so that higher values would consistently reflect worse pathology across all biomarkers. This transformation was applied to facilitate interpretability in the figures. To avoid

confusion, we have now removed the “(*-1)” notation from the legend and clarified the transformation in the Methods section.

Comment 9. Why were HCV and CSF A β 1-42 values inverted? The correlation coefficients are negative for A β 1-42 (as you would expect) suggesting they were not inverted. The legend needs revising for clarity. Same for Supplementary Figure 1 legend.

Response: Thank you for the attention to this point. We inverted HCV and CSF A β 42 values to align with the logic of other biomarkers, where higher values indicate greater pathology. This adjustment was made to facilitate interpretation, as increased values of HCV and A β 42 typically reflect less pathology. We have reviewed the data and we confirm that the values for HCV and CSF A β 42 were indeed inverted. This inversion is consistent with the expected pattern of increasing tau PET SUVR in the meta-temporal ROI, and the negative coefficients are only in reference to CSF t-tau, Ng, and SNAP25, in the CU group, which highlights the similarities of synaptic biomarkers with CSF t-tau. The correlation between higher levels of CSF t-tau, synaptic markers and lower A β burden in CU individuals has been demonstrated in several studies (Kester et al., 2015; Mielke et al., 2019; Tible et al., 2020), but not in all (Galasko et al., 2019), concerning t-tau. Additionally, the direction of the relationship between synaptic biomarker and A β burden in CU individuals appears to vary depending on amyloid-positivity, with the significance of the correlation dependent on tau pathology (CSF t-tau and p-tau181) (Milà-Alomà et al., 2021). In our study, 65% of CU individuals were A β -negative, which may help explain the direction of the correlation observed. We now have addressed this in the discussion (page 11 line 11) as well:

“We found an association between CSF t-tau and CSF A β 42, further supporting the notion that CSF t-tau may be sensitive to AD pathology.^{18,19} However, the modest associations of CSF t-tau with tau-PET, highlight it is not a strong proxy for tau tangle pathology^{17,41}. Our findings also suggest that higher levels of CSF t-tau, Ng and SNAP25 are associated with lower A β burden in CU individuals, consistent with some studies,^{16,23} but not all.²⁴”

We can also see the direction of the associations - before inversion and log-transformation - between different biomarkers and CSF A β 42 in the figure below:

Comment 10. There are errors in Figure 4. and supplementary Figure 3. The last two categories are both labelled as N+/S+.

Response: We appreciate the attention to detail in our manuscript. The labels were corrected in Figure 4 and Supplemental Figure 6.

Figure 4. CSF t-tau is increased in individuals with abnormal synaptic degeneration regardless of concomitant presence of neurodegeneration. (a) Bar graphs show the distribution of CU, MCI and dementia across synaptic degeneration (S) and neurodegeneration (N) groups. N positivity was based on HCV and S positivity was based on CSF Ng (N⁻S⁻: n = 85; N⁻S⁺: n = 137; N⁺S⁻: n = 110; S⁺N⁺: n = 252). Cutoffs were calculated as above or below median from CU A β ⁻ individuals. (b) Violin plots show CSF t-tau levels in individuals with abnormal S and/or N in the whole population (CU and CI) across all cohorts. Linear regression model with dummy variables, adjusted for age, sex, cognitive status, A β burden and cohort was employed to compare CSF t-tau levels across groups. * P-value < 0.05. Amyloid- β (A β). Cognitively unimpaired (CU). Hippocampal volume (HCV). Mild cognitive impairment (MCI). Neurodegeneration (N). Neurogranin (Ng). Synaptic degeneration (S).

Minor issues.

Comment 11. Information is lacking in the Study population section relating to which ADNI phase was used and download date.

Response: Thank you for the comment. We added the information in the Methods (page 14 line 15):

“The data for this study were collected from ADNI phases ADNI1, ADNIGO, ADNI2, and ADNI3, with the most recent data downloaded in April 2024.”

Comment 12. Participants were considered S+ based on either Ng or SNAP25 values above median from CU A β -negative individuals.⁵⁰ I do not think ref 50 is correct here.

I have spotted several references that do not seem to be correct (eg., 33, 34, 50?). This should be revised.

Response: We thank the reviewer for noting these important discrepancies. Indeed, there were several references that were not matching the intended publications as well as some duplicates. They are now all correct.

Comment 13. P8. Line 182. “Overall, we observed weak correlations between neurodegeneration synaptic biomarkers.” Should read “Overall, we observed weak correlations between neurodegeneration and synaptic biomarkers”

Response: We value the attention, and we have corrected to:

“Overall, we observed weak correlations between neurodegeneration and synaptic biomarkers”.

Comment 14. In the discussion Page 2 line 254. There is repetition of previous paragraph

Response: Thank you for your observation. We have significantly revised the discussion to eliminate redundancy and improve the flow of ideas (page 9 line 2):

“In this study, we investigated the association of CSF t-tau with markers of synaptic and axonal/neuronal degeneration across the aging and the AD spectrum in two independent large cohorts. Our findings suggest that CSF t-tau was more closely associated with classical biomarkers of synaptic degeneration than those of neuronal degeneration. Notably, we found evidence that abnormalities in biomarkers of synaptic degeneration, even in the absence of neuronal degeneration abnormality indexed by HCV, were strongly associated with increased CSF t-tau levels. Thus, CSF t-tau shows the potential to be used as a proxy of synaptic degeneration.

We found strong associations between CSF t-tau with CSF Ng and SNAP25 in both CU and CI individuals, reinforcing mounting evidence showing a strong link between CSF t-tau and Ng^{15,16,24–27} SNAP25,^{24,25} GAP43,^{24,25} synaptotagmin-1,²⁵ and neuropentraxin-2.²⁴ In CI individuals, we observed a modest association between CSF t-tau and HCV, CSF NfL and FDG-PET, while in CU individuals, CSF t-tau was only moderately associated with CSF NfL. These results align with prior studies in CI individuals showing low-moderate associations with FDG-PET,²⁸ HCV and cortical thickness^{28–30} and plasma NfL,³⁰ but differ from reports that found significant associations in the CU population³¹ or observed stronger correlations in a CU and CI combined sample.^{32,33} Given the limited evidence in CU populations and variability across studies, further research is needed to clarify t-tau’s role in preclinical stages. Altogether, these findings suggest that CSF t-tau is well associated with synaptic biomarkers before cognitive impairment, but only moderately - and most frequently only in symptomatic individuals - with measures of atrophy, neuronal injury and glucose uptake.”